# Twisting of the zebrafish heart tube during cardiac looping is a *tbx5*-dependent and tissue-intrinsic process

Federico Tessadori[1]*, Erika Tsingos[2], Enrico Sandro Colizzi[2,3], Fabian Kruse[1], Susanne C van den Brink[1], Malou van den Boogaard[4], Vincent M Christoffels[4], Roeland MH Merks[2,3,5], Jeroen Bakkers[1,6]*

[1]Hubrecht Institute-KNAW and University Medical Center Utrecht, Utrecht, Netherlands; [2]Mathematical Institute, Leiden University, Leiden, Netherlands; [3]Origins Center, Leiden University, Leiden, Netherlands; [4]Amsterdam UMC, University of Amsterdam, Department of Medical Biology, Amsterdam Cardiovascular Sciences, Amsterdam, Netherlands; [5]Institute of Biology, Leiden University, Leiden, Netherlands; [6]Department of Pediatric Cardiology, Division of Pediatrics, University Medical Center Utrecht, Utrecht, Netherlands

**Abstract** Organ laterality refers to the left-right asymmetry in disposition and conformation of internal organs and is established during embryogenesis. The heart is the first organ to display visible left-right asymmetries through its left-sided positioning and rightward looping. Here, we present a new zebrafish loss-of-function allele for *tbx5a*, which displays defective rightward cardiac looping morphogenesis. By mapping individual cardiomyocyte behavior during cardiac looping, we establish that ventricular and atrial cardiomyocytes rearrange in distinct directions. As a consequence, the cardiac chambers twist around the atrioventricular canal resulting in torsion of the heart tube, which is compromised in *tbx5a* mutants. Pharmacological treatment and ex vivo culture establishes that the cardiac twisting depends on intrinsic mechanisms and is independent from cardiac growth. Furthermore, genetic experiments indicate that looping requires proper tissue patterning. We conclude that cardiac looping involves twisting of the chambers around the atrioventricular canal, which requires correct tissue patterning by Tbx5a.

**\*For correspondence:**
f.tessadori@hubrecht.eu (FT);
j.bakkers@hubrecht.eu (JB)

**Competing interests:** The authors declare that no competing interests exist.

## Introduction

Bilateral animals such as vertebrates, while being symmetric on the outside when divided through the sagittal plane, have left-right (LR) asymmetrically arranged internal organs. LR asymmetry of organ disposition and form supports proper development and function of the organism throughout life.

The embryonic heart is the first organ to visibly break LR symmetry of the vertebrate embryo (*Desgrange et al., 2018* and references therein). The heart starts out as a linear tube positioned at the midline, which subsequently bends toward the right, initiating an ensemble of developmentally regulated complex processes referred to as cardiac looping (*Patten, 1922*). The looped heart tube is either a flat S-shape in fish or a helix in amniotes (chick and mouse) (*Desgrange et al., 2018*). Correct looping is closely intertwined to proper patterning and alignment of the inflow and outflow tracts, cardiac chambers and atrioventricular canal, which are crucial to establish and maintain heart function. Indeed, cardiac looping defects in humans can result in severe congenital heart defects such as transposition of the great arteries (TGA), double outlet right ventricle (DORV), and Tetralogy of Fallot (TOF) (*Lin et al., 2014*).

Correct cardiac looping depends on both tissue intrinsic and extrinsic mechanisms. Establishment of LR asymmetry involves an extrinsic mechanism that influences cardiac looping. In most vertebrates, this LR asymmetry is established during embryogenesis due to the activity of the LR organizer, called the node in mice and Kupffer's vesicle in zebrafish. The LR organizer is a transient structure consisting of ciliated cells, located in the posterior part of the embryo (*Essner et al., 2002*). Rotation of the cilia results in a directed fluid flow (nodal flow), which breaks the symmetry by inducing left-sided-specific expression of Nodal and Pitx2 (*Meno et al., 1998*; *Okada et al., 1999*). Left-sided Nodal expression regulates the asymmetric position and dextral looping of the heart (*Meno et al., 1998*; *Baker et al., 2008*; *Long et al., 2003*; *Noël et al., 2013*; *Levin et al., 1997*). In zebrafish, LR symmetry is first broken when the linear heart tube arises from an initial flat disc between 20 and 24 hr post-fertilization (hpf; reviewed in *Stainier, 2001*). As its formation progresses, the inflow pole moves to the left side of the midline in a process referred to as cardiac jogging (*Chen et al., 1997*). This breaking of LR symmetry is dependent on left-sided Nodal expression (*Long et al., 2003*; *Grimes et al., 2020*; *Montague et al., 2018*). After this, the heart tube undergoes cardiac looping, which under normal conditions is dextral (rightward). If the function of the LR organizer is affected, a sinistral (leftward) loop can be observed (*Noël et al., 2013*; *Noël et al., 2016*). Based on mutant analysis, it was suggested that cardiac jogging can be separated from cardiac looping and that there are likely separate mechanisms that regulate these processes (reviewed by *Bakkers et al., 2009*). Corroborating such a model, we previously demonstrated that while left-sided Nodal expression directs cardiac jogging, a separate, tissue-intrinsic mechanism drives looping morphogenesis (*Noël et al., 2013*). Intrinsic LR asymmetry has been observed in various tissues and organs of invertebrates (reviewed in *Inaki et al., 2016*). In *Drosophila*, the hindgut and the genitalia show LR asymmetry (*Sato et al., 2015*; *Taniguchi et al., 2011*), for which myosin seems to be the major determinant (*Hozumi et al., 2006*; *Lebreton et al., 2018*). LR asymmetry is not only observed at the organ and tissue level, but also in single cells (reviewed in *Pohl, 2015*). For example, human leukemia cells preferentially polarize to the left of an imaginary axis between the nucleus and the centrosome (*Xu et al., 2007*). The actin cytoskeleton and actomyosin interactions are important for the observed intrinsic chirality of cells (reviewed in *Satir, 2016*) as chiral actin cytoskeletal organization was observed in cells on micropatterns (*Tee et al., 2015*; *Wan et al., 2011*). As cardiomyocytes display LR asymmetries during cardiac looping, and heart looping morphogenesis requires actomyosin activity, this presents the exciting hypothesis that vertebrate heart looping depends on tissue- and cell-intrinsic chirality (*Noël et al., 2013*; *Merks et al., 2018*; *Ray et al., 2018*).

To identify novel factors and mechanisms that drive cardiac looping, we have performed forward genetic screens in zebrafish (*Noël et al., 2013*; *Smith et al., 2011a*; *Tessadori et al., 2015*; *Wienholds et al., 2003*). In such a screen we identified the *oudegracht* (*oug*) mutant in which cardiac jogging was unaffected while cardiac looping was compromised. We found that a novel loss-of-function allele for *tbx5a*, one of the two zebrafish paralogues of *Tbx5*, was responsible for the cardiac looping defect in *oug* mutants. Tbx5 is a transcription factor which acts as a master regulator of cardiac development, with established roles in cardiomyocyte differentiation, conduction system development, and septation across vertebrates, including humans (*Jensen et al., 2013*; *Mori and Bruneau, 2004*); however, a link to intrinsic heart looping morphogenesis has not been established yet. To gain a better understanding of cardiac looping, we performed live two-photon confocal imaging in wild type and *oug* mutant embryos and mapped cardiomyocyte behavior at a single-cell level. Our study establishes that during looping, cardiomyocytes in the forming ventricle and atrium actually rearrange toward the outer curvatures of the chambers. Hence, the ventricle and the atrium undergo asymmetric rotational movements around the atrioventricular canal, effectively transmitting a twisting transformation to the heart tube, a process which we show to be defective in $tbx5a^{-/-}$ zebrafish mutants. To address which processes exert a regulatory role in this major cellular rearrangement, we manipulated cardiac looping by chemical treatment or ex vivo culture and analyzed single-cell behavior during heart morphogenesis. Finally, rescue of the $tbx5a^{-/-}$ cardiac phenotype in a $tbx2b^{-/-}$ background establishes that the intrinsic looping morphogenesis relies on correct genetic patterning during cardiac development.

## Results

### Tbx5a is required for cardiac looping and patterning

We have performed several forward genetic screens to identify genes that regulate LR patterning and heart looping morphogenesis (*Noël et al., 2013*; *Smith et al., 2011a*; *Tessadori et al., 2015*; *Wienholds et al., 2003*). In short, embryos were screened around 28 hpf for correct formation and asymmetry of the cardiac tube, and at 50 hpf to assess cardiac looping. In one of these screens, the recessive and lethal *oudegracht (oug)* mutation was identified, named after the stretched S-shaped canal in the city centre of Utrecht (NL). The *oug* mutants displayed cardiac edema, defective cardiac looping at 50 hpf (*Figure 1A–C*) and reduced heartbeat rate (not shown). LR patterning was unaffected in *oug* embryos since the direction of cardiac jogging was predominantly leftward and the laterality of the visceral organs was not affected (*Figure 1C*). Morphologically, *oug* mutants grow normally, although importantly they lack development of the pectoral fin buds (*Figure 1B*). Using positional cloning and direct sequencing, we determined that *oug* mutants carry a point mutation resulting in a premature truncation of the Tbx5a transcription factor (*Figure 1D–F*; ENS-DARG00000024894). The *oug* mutation is a recessive, fully phenotypically penetrant mutation as crossing of heterozygous *oug* carriers yielded approximately 25% progeny displaying a cardiac looping defect and absence of fin buds (*Figure 1G*), conforming to the corresponding Mendelian inheritance pattern. To confirm that *oug* affects the *tbx5a* locus (NM_130915), we carried out a complementation test with a previously identified *tbx5a* mutant allele, *heartstrings (hst)* which was also reported to display cardiac looping and fin bud formation defects (*Garrity et al., 2002*). Crossing of heterozygous *oug* and *hst* carriers yielded about 25% embryos in which both of these phenotypes were present, thereby confirming that the heart and fin phenotypes observed in *oug* embryos are caused by a mutation in *tbx5a* (*Figure 1G*).

Embryos homozygous for the *oug/tbx5a* allele display consistent reduced dextral looping (*Figure 1—figure supplement 1*), especially noticeable when compared to the relative variability in the looping defect of *hst* (*Figure 1—figure supplement 1*).

As *tbx5a* is expressed throughout the myocardium (*Figure 1H,I*), where it regulates patterning of the heart in chamber (working) and AV canal (non-working) myocardium we performed in situ hybridization (ISH) using markers for the AV canal and chamber myocardium. In agreement with such a role for Tbx5 we observed in *oug/tbx5a* mutants a strong reduction in chamber differentiation (*nppa*, *Figure 1H*) while the AV canal region was expanded as revealed by expanded domains of expression for *bmp4* and *tbx2b* (*Figure 1H*). The latter contrasted with *hst/tbx5a* mutant AV canals in which *tbx2b* transcripts were just-detectable (*Figure 1H*) or reported to be absent (*Garrity et al., 2002*). In accordance with our observations on the AV canal myocardium, we also detected increased expression of the AV endocardial markers *has2* and *versican* (*Figure 1H*).

The *oug/tbx5a* allele (hereafter, and throughout the manuscript referred to as *oug*) truncates Tbx5a at amino acid 147 (out of 492; *Figure 1F*), resulting in the loss of approximately 50% its DNA-binding T-box domain, which is crucial for its function (*Wilson and Conlon, 2002*). This is not the case for the *hst/tbx5a* allele, which does not affect the T-box domain (*Figure 1F*).

To address whether the difference in AV canal phenotype (i.e. expression of *tbx2b*) between *oug* and *hst* mutants could be due to differences in activity of the perspective Tbx5a mutations, we carried out an in vitro test for Tbx5a activity (*Figure 1J*). Tbx5 activity was measured using a regulatory region of the *nppa* gene that contains a T-box-binding site driving luciferase expression. Our results show that while Tbx5a with the *oug* mutation causes an almost complete loss of luciferase expression, Tbx5a with the *hst* mutation retained a significantly higher capacity to induce luciferase expression (*Figure 1J*). Hence, the defect in cardiac gene patterning and accompanying failure to complete cardiac looping in *oug* mutant embryos are the result of loss of Tbx5a function.

### Time-lapse imaging reveals twisting of the chambers around the AV canal

Cardiac looping in zebrafish can be observed from 28 hpf and is considered to be completed, including chamber ballooning, at around 55 hpf. During this process, the heart tube not only changes position with respect to the overall geometry of the embryo (*Figure 2—figure supplement 1*) but also seemingly undergoes flat bending (or planar buckling) along its anterior-posterior axis

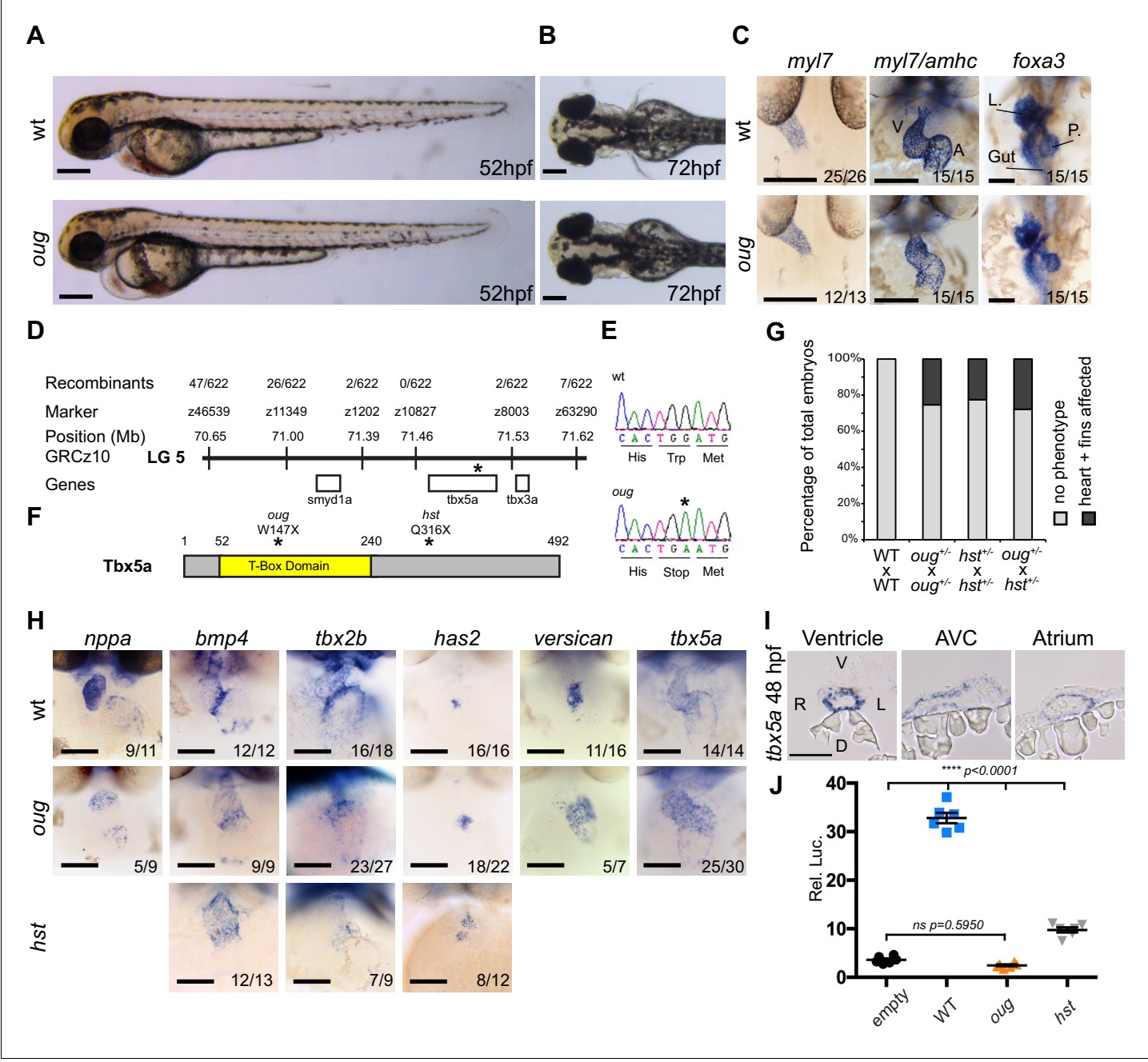

**Figure 1.** The *oudegracht (oug)* mutant carries a *tbx5a* null allele and displays defective cardiac looping. (**A**) Lateral view of wt and *oug* mutant embryos at 52 hpf. Note the cardiac edema in *oug*. (**B**) At 72 hpf dorsal observation of *oug* mutant embryos reveals absence of lateral fins. (**C**) Two dpf *oug* mutant embryos display defective cardiac looping but normal asymmetric positioning of the internal organs. L, liver; P, pancreas (**D**) Mapping and genomic position of the *oug* mutation (indicated by the asterisk). (**E**) A single-nucleotide substitution in *tbx5a* (G to A) resulting in a tryptophan (Trp; TGG) to stop (TGA) mutation segregates with the *oug* phenotype. (**F**) Tbx5a is truncated at amino acid 147 in *oug,* in its T-Box domain. The *hst* allele (Q316X; *Garrity et al., 2002*) is included for comparison (**G**) Complementation test. Outcross of *oug*[+/-] to *hst*[+/-] fails to complement the *oug* cardiac and pectoral fin bud phenotype. (**H**) Gene patterning is affected in *oug* hearts at 2dpf. Expression of *nppa* is reduced in the cardiac chambers while expression of *bmp4* and *tbx2b* is expanded in the AV canal. Cardiac cushion markers *has2* and *versican* also show expanded expression domains. ISH for *hst* is shown for comparison: while *bmp4* and *has2* display expanded expression domains as in *oug*, *tbx2b* is barely detectable. Transcripts for *tbx5a* are detected in wt and *oug* mutants. (**I**) Transcripts for *tbx5a* can be detected evenly in transversal sections through the entire 2 dpf heart tube. (**J**) Luciferase assay establishes that *oug* retains virtually no activity. Mean values ± SEM are shown. Scale bars (**A,B,C,H**): 100 μm; (**I**): 50 μm.

The online version of this article includes the following source data and figure supplement(s) for figure 1:

**Source data 1.** Source files for data presented in panels G and J.

*Figure 1 continued on next page*

*Figure 1 continued*

**Figure supplement 1.** Variability of the looping phenotype in *oug* and *hst* zebrafish alleles of *tbx5a*.

(*Figure 2—figure supplement 1*). To get more insight into this transformation, we have defined a left-right and a superior-inferior axis of the heart tube at 28 hpf (*Figure 2A*) and we followed the movements of individual cardiomyocytes approximately from 28 hpf to 38 hpf (*Figure 2A*; *Figure 2—figure supplement 2*; *Figure 2—video 1*) in hearts in which cardiac contractions were suppressed (*Sehnert et al., 2002*). At this early stage, the embryonic zebrafish heart displays normal heart morphogenesis in the absence of heartbeat (*Noël et al., 2013*). Individual cardiomyocytes were tracked (*Figure 2B*) and the start and end point of each trace was used to obtain the individual track displacement, hence quantifying the displacement of each tracked cardiomyocyte and representing it as a vector (*Figure 2C*; *Figure 2—figure supplement 2*; *Figure 2—video 2*). Based on the starting location at the beginning of their corresponding track, cardiomyocytes were categorized in three regions: ventricle, atrium, and AV canal (*Figure 2D*). Visual inspection of these 'displacement maps' revealed coherent cellular movements within the heart chambers (*Figure 2E–F*). Comparison of the displacement tracks in the superior and inferior sides of the heart tube revealed large differences. Most strikingly, the vectors in the superior and inferior sides of the atrium pointed in different directions (*Figure 2E–F*). If planar buckling was the principal contributor to the transformation, the expected displacement vectors for the superior and inferior sides of each chamber would be similar. Instead, in the atrium these vectors pointing in near opposite directions suggested that the atrium rotates during cardiac looping. This impression was corroborated by the presence of cardiomyocyte tracks with major Z-displacement at the outer (*Figure 2G*; asterisks) and inner (*Figure 2H*; arrowheads) curvatures of the atrium, both compatible with a rotational transformation of the chamber. To more precisely quantify rotation of the cardiac chambers, we subjected all time-lapse movies to the following procedure: first, we stabilized residual drift of the heart tube by rooting the centroid (for definition see Appendix 1-Supplementary Methods) of the AV canal at the origin (0,0,0) of the coordinate system throughout all timepoints (*Figure 2I*). Second, we identified two axes: the first running from the AV canal centroid to the centroid of the ventricle, the other running from the AV canal to the centroid of the atrium. For each timepoint, we unfolded the axis by rotating the positions of the entire atrium and ventricle, with the AV canal acting as a 'hinge' rooted at the origin, to make the axes overlap with their respective position at the start of the timelapse (*Figure 2I'*). After this 'computational unfolding' only the rotation of the cardiomyocytes around either the atrium axis or the ventricle axis remained in the dataset. Third, to quantify this rotation, we measured the angle α subtended between the starting and ending cellular positions at consecutive time points (*Figure 2I''*; *Figure 2—video 3*). The rotational velocity ω of the cells is given by this angle divided by the time Δt between two timepoints (Supplementary *Equation 17* in Supplementary Methods). By integrating the average of all cells' rotational velocity to time (i.e. cumulative addition of the average rotation angles at consecutive timepoints to obtain the total angle traveled), we obtain the rotation of each chamber around each of the axes (*Figure 2J*; for detailed explanation see Appendix 1-Supplementary Methods). We observed that the absolute value of the average total rotation steadily increases for both the ventricle and the atrium in all hearts (n = 5), with clearly separating values for the ventricle (negative) and atrium (positive) (*Figure 2J*), indicating that the chambers rotate in opposite directions. Values for cells in the AV canal displayed a much more erratic behavior, with variability in positive and negative total rotation angle values between and within the tracks (*Figure 2—figure supplement 2*). During cardiac looping, the angular velocities of the ventricle (negative) and atrium (positive) differ consistently from one another (*Figure 2K*), while the AV canal hardly rotates (*Figure 2—figure supplement 2*). Altogether these observations show that rotation of the ventricle and the atrium in opposite directions around the AV canal twists the heart tube during development.

## Genetic tracing of left and right cardiac fields reveals twisting of the cardiac tube

During linear heart tube formation the cardiac disc rotates in a clockwise direction (from a dorsal view), while at the same time invagination of the right- and posterior sides results in a three-

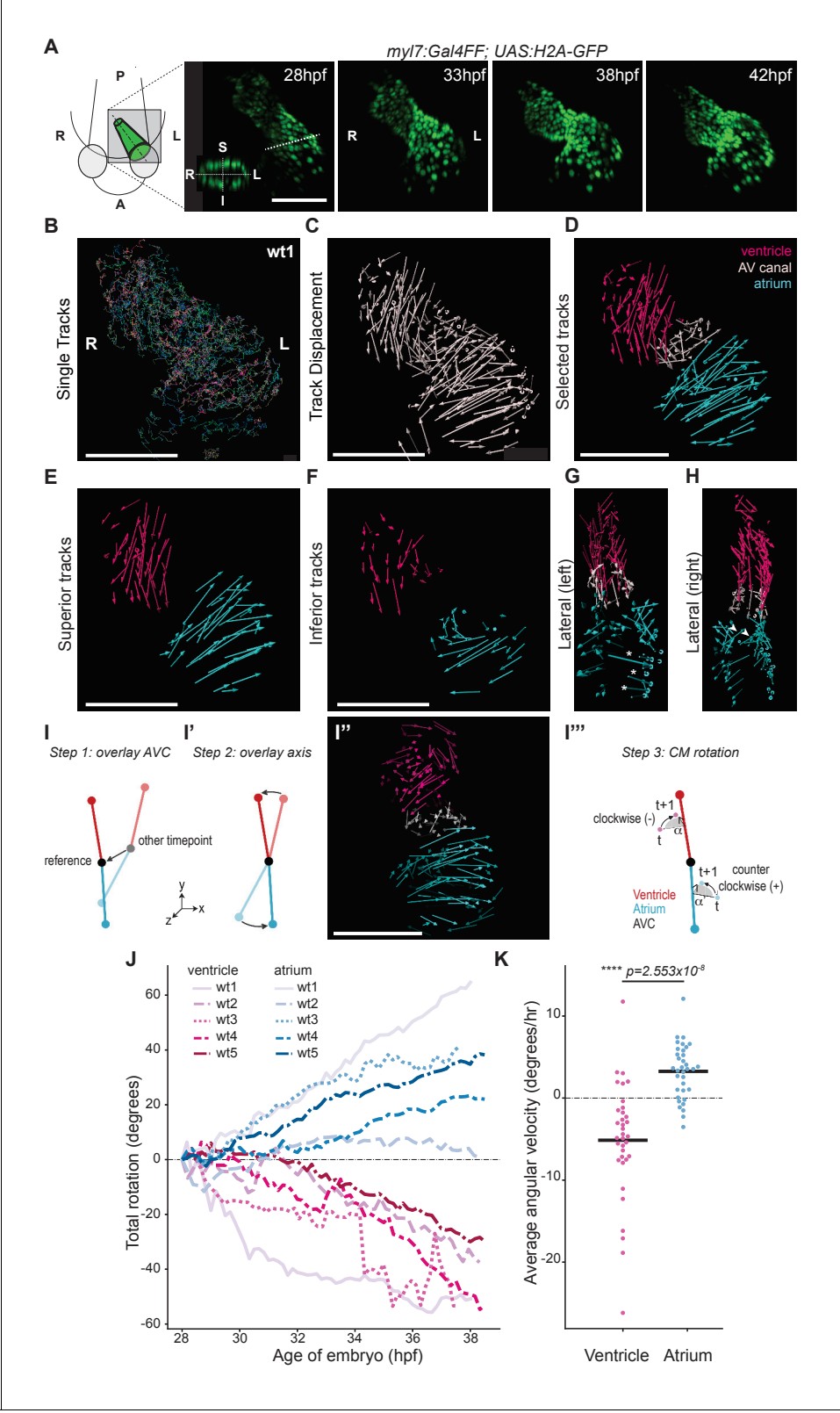

**Figure 2.** Cardiac looping is accompanied by opposite rotation of the cardiac chambers. (A) Time-lapse imaging is carried out on *tg(myl7:Gal4FF; UAS: H2A-GFP)* embryos. In the 28 hpf panel, the dashed line indicates the position of the transversal section shown in the bottom left corner, in which the superior (S), inferior (I), right (R) and left (L) sides of the heart tube are defined. One representative heart is shown. A: anterior; P: posterior. (B) Total tracks (Ventral View). Each track is color-coded and is assigned an ID number. (C) Track displacement vectors for each single trace. (D) Track

*Figure 2 continued on next page*

*Figure 2 continued*

displacement vectors to be analyzed are selected, categorized by visual inspection and color-labeled accordingly. (E) Cardiac displacement vectors on the superior side of the ventricle and atrium and (F) on the inferior side of the cardiac chambers. (G) Displacement of cardiomyocytes at the outer curvature (asterisks) and (H) at the inner curvature (arrowheads) of the atrium are compatible with rotation of the chamber. (I–I''') Computational unfolding and angular velocity measurement. (I-I'') Steps 1 and 2 (I, I') taken to computationally unfold the heart tube, resulting in the vector map shown in I''. The angular velocity of the cardiomyocytes is then calculated in the plane perpendicular to the axis (I'''). A detailed description of the methodology is available in the SI (J) Cumulative rotation angle for the ventricle (shades of red) and atrium (shades of blue) in wild-type hearts. Note the opposite direction of rotation of the two chambers. Positive values represent anti-clockwise rotation and negative values represent clockwise rotation with respect to the outflow of the heart. (K) Comparison of the average angular velocity for each replicate per 1.5 hr time window displayed by the chambers analyzed in (J). Horizontal bars: mean values. Scale bars: 100 μm.

The online version of this article includes the following video, source data, and figure supplement(s) for figure 2:

**Source data 1.** Source files for data presented in panels J and K.

**Figure supplement 1.** Zebrafish cardiac looping.

**Figure supplement 2.** Analyzed wt hearts and AV canal analysis.

**Figure supplement 2—source data 1.** Source files for data presented in panel B of *Figure 2—figure supplement 2*.

**Figure 2—video 1.** Timelapse of wt1 *tg(myl7:Gal4FF; UAS:H2A-GFP)* heart.
https://elifesciences.org/articles/61733#fig2video1

**Figure 2—video 2.** Displacement vectors for wt1 heart (360° rotation).
https://elifesciences.org/articles/61733#fig2video2

**Figure 2—video 3.** Computational processing of wt1 heart timelapse.
https://elifesciences.org/articles/61733#fig2video3

**Figure 2—video 4.** Timelapse of wt2 *tg(myl7:Gal4FF; UAS:H2A-GFP)* heart.
https://elifesciences.org/articles/61733#fig2video4

**Figure 2—video 5.** Timelapse of wt3 *tg(myl7:Gal4FF; UAS:H2A-GFP)* heart.
https://elifesciences.org/articles/61733#fig2video5

**Figure 2—video 6.** Timelapse of wt4 *tg(myl7:Gal4FF; UAS:H2A-GFP)* heart.
https://elifesciences.org/articles/61733#fig2video6

**Figure 2—video 7.** Timelapse of wt5 *tg(myl7:Gal4FF; UAS:H2A-GFP)* heart.
https://elifesciences.org/articles/61733#fig2video7

**Figure 2—video 8.** Displacement vectors for wt2 heart (360° rotation).
https://elifesciences.org/articles/61733#fig2video8

**Figure 2—video 9.** Displacement vectors for wt3 heart (360° rotation).
https://elifesciences.org/articles/61733#fig2video9

**Figure 2—video 10.** Displacement vectors for wt4 heart (360° rotation).
https://elifesciences.org/articles/61733#fig2video10

**Figure 2—video 11.** Displacement vectors for wt5 heart (360° rotation).
https://elifesciences.org/articles/61733#fig2video11

**Figure 2—video 12.** Computational processing of wt2 heart timelapse.
https://elifesciences.org/articles/61733#fig2video12

**Figure 2—video 13.** Computational processing of wt3 heart timelapse.
https://elifesciences.org/articles/61733#fig2video13

**Figure 2—video 14.** Computational processing of wt4 heart timelapse.
https://elifesciences.org/articles/61733#fig2video14

**Figure 2—video 15.** Computational processing of wt5 heart timelapse.
https://elifesciences.org/articles/61733#fig2video15

dimensional cone (*Baker et al., 2008*; *Rohr et al., 2008*; *Smith et al., 2008*; *de Campos-Baptista et al., 2008*). As a consequence of this rotation and folding, the cardiomyocytes originating from the left cardiac field form the superior side of the tube, while cells originating from the right cardiac field form the inferior side at approximately 24 hpf (*Bakkers et al., 2009*). A model has been proposed in which this clockwise rotation is followed by a counterclockwise rotation just before or during looping, which would restore the original left-right orientation of the cardiac cells (*Baker et al., 2008*). This two-rotation model would not be compatible with our observations from the cell tracking of ventricular cardiomyocytes. In an attempt to resolve this, we generated a new transgenic line that would allow an accurate tracing of cells derived from the left and right cardiac fields. The transgenic line, referred to as *tg(0.2lntr1spaw:GFP)* (*Figure 3—figure supplement 1*) was made by using a

highly conserved 0.2 kb sequence in the first intron of the Nodal-related gene *spaw*, which acts as an asymmetric enhancer (ASE; *Fan et al., 2007*; *Norris and Robertson, 1999*). This ASE sequence drives GFP expression in the left lateral plate mesoderm (LPM) during somatogenesis. While *spaw* mRNA is no longer detectable in the left heart field beyond 30 hpf, the stability of the fluorescent protein allows us to follow left-derived GFP-positive cells up to 2 dpf. This line could therefore be used in combination with a *myl7* fluorescent reporter to trace cells originating from the left and right cardiac fields during cardiac looping stages and address how these cells behave during cardiac looping morphogenesis.

We first wanted to test whether we could confirm the clockwise rotation during linear heart tube formation, which results in left-derived cells occupying the superior side and right-originating cells occupying the inferior side of the tube (*Rohr et al., 2008*; *Smith et al., 2008*). Indeed, this clockwise rotation is also observed in vivo, in *tg(myl7:Gal4FF; UAS:RFP; 0.2Intr1spaw:GFP)* zebrafish embryos as localization of *0.2Intr1spaw:GFP* expressing cells is confined to the superior side of the tube (*Figure 3A,A'*). We then proceeded to use these transgenic lines to analyze the localization of the left- and right-originating cells in the looped heart. Interestingly, at this stage, left-originating cells localizing to the superior side of the heart tube are now located ventrally with respect to the inferior side of the heart tube, which is due to an extension of the embryo and a 180 degrees flip of the heart tube (*Figure 3B* and *Figure 2—figure supplement 1*). In addition, in cross-sections we observed left-originating cells at the outer curvatures of both chambers, reaching, especially visible in the ventricle, the inferior side of the heart (*Figure 3B*, arrowheads). Concomitantly, the region at the inner curvature of the atrium is only RFP-positive, indicating the right origin of these cardiomyocytes (*Figure 3B*). To confirm these observations, we used an additional reporter line in which the regulatory sequences of the *lefty2* gene drive expression of Gal4FF (*Asakawa et al., 2008*), referred to as *tg(lft2BAC:Gal4FF)*(*Derrick et al., 2021*). This line, when combined with a UAS fluorescent reporter line, recapitulated endogenous *lefty2* expression in the cardiac disc (*Figure 3—figure supplement 2*). Analysis of the localization of the left- and right-originating cells in the looped heart in *tg(lft2BAC:Gal4FF)* by fluorescence immunolabeling (*Figure 3—figure supplement 2*) corroborated our results obtained with *tg(0.2Intr1spaw:GFP)*. Together, these observations are consistent with those from our time-lapse imaging and cell tracing. Furthermore, they confirm our conclusion that cardiac chambers twist around the AV canal in opposing directions resulting in torsion of the heart tube.

## Tbx5a is required for the twisting of the cardiac chambers

To address the role of Tbx5a in the observed twisting of the cardiac chambers, we first crossed the *oug* mutation into the *tg(myl7:Gal4FF; UAS:RFP; 0.2Intr1spaw:GFP)*. Contrary to observations in wild-type hearts, we observed that the outer curvature of both the ventricle and atrium in *oug* mutant hearts are largely devoid of left-originating GFP+ cells (*Figure 3C*). In transversal sections of the ventricle, left-originating cells remain largely localized to the superior side of the heart tube (*Figure 3C*). The domain occupied by left-originating cells remained virtually unchanged when compared to the situation at the end of cardiac jogging, suggesting a lack of twisting and the absence of torsion in hearts lacking Tbx5a.

Next, we time-lapsed and analyzed cardiomyocyte displacements in five *oug* mutant embryos in the same manner as we did for siblings using the *tg(myl7:Gal4FF; UAS:H2A-GFP)* line (*Figure 4A–E*; *Figure 4—figure supplement 1*; *Figure 4—videos 1–3*). Cardiomyocyte tracks on the superior and inferior sides of the cardiac chambers did not display the visible difference in rotation direction (*Figure 4D–E*) that was observed in the wild-type situation. Moreover, we did not observe major retreating or advancing Z-displacements at the outer and inner curvature, respectively (*Figure 4F–G*). This suggests that, while some bending of the cardiac tube happens during cardiac looping in *oug/tbx5a*, rotation of the chambers is strongly reduced if present at all. Plotting the average total rotation angle for the mutant ventricles and atria (*Figure 4H*), did not result in a clear separation of the tracks for each chamber type, as was the case for the wild type (compare with *Figure 2J*). Many of the tracks successively display positive and negative rotation angle values, which would indicate that during the time-lapse acquisition time, there is little concerted movement of the cardiomyocytes in the chambers. Furthermore, the absence of separation of the ventricular and atrial tracks indicates that the twisting of the heart tube (i.e. the opposite rotation of atrium and ventricle) is largely absent in *oug*. Comparison of the mean ventricular and atrial angular velocity values yielded no significant

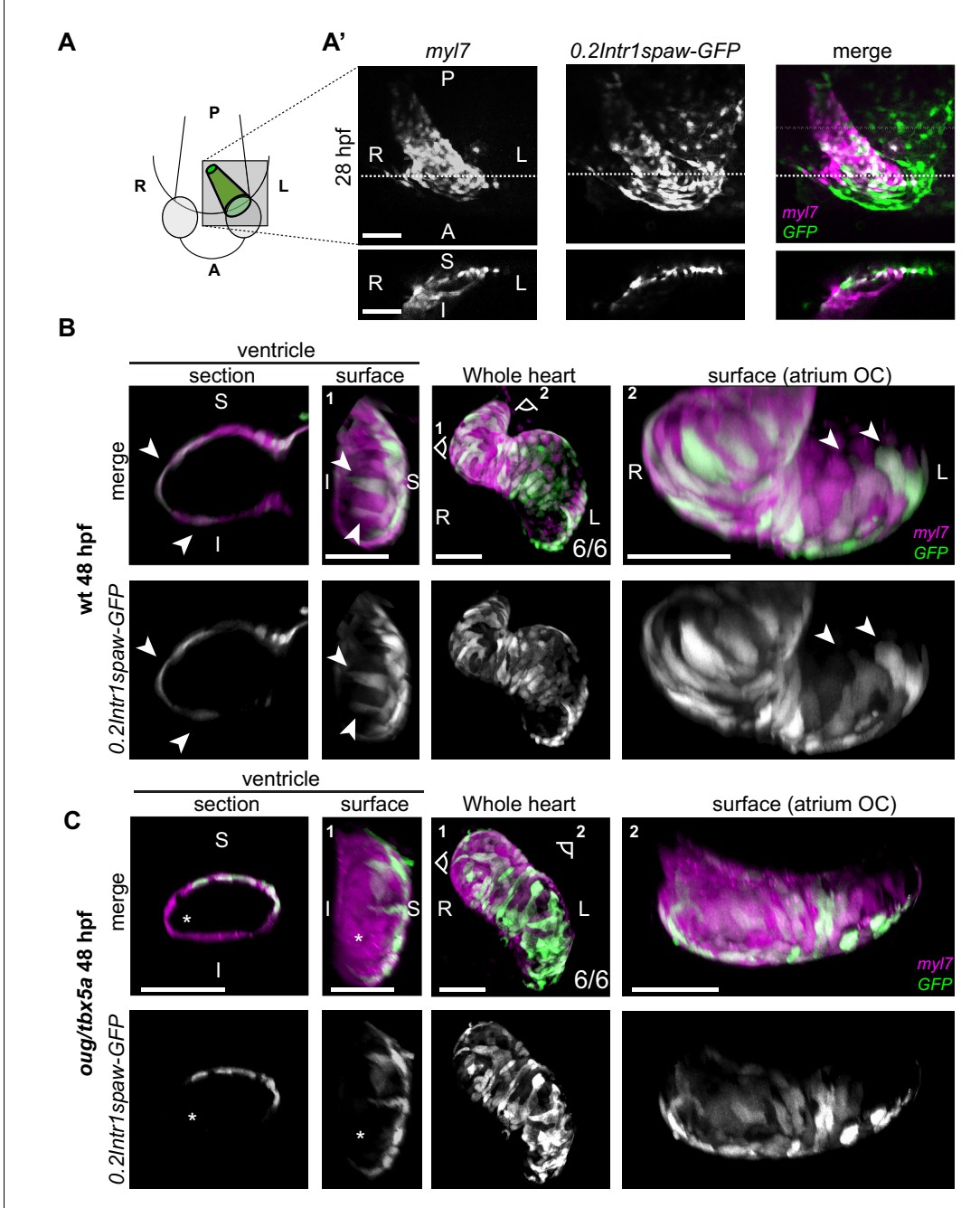

**Figure 3.** Origin and final positioning of left- and right-originating cardiomyocytes during cardiac looping. (**A**) At 28 hpf, as cardiac jogging towards the anterior left side of the embryo is completed, (**A′**) the *tg(0.2Intr1spaw:GFP)* labels cardiomyocytes localizing to the superior side of the cardiac tube (section). (**B**) By 48 hpf cardiac looping morphogenesis is accompanied by displacement in opposite directions of left-originating cardiomyocytes toward the outer curvatures of the ventricle and the atrium (arrowheads in the section and surface view panels). (**C**) At 48 hpf, the *oug* mutant heart tube fails to display any constriction at the AV canal and left-originating cardiomyocytes are not visible in the region around the outer curvatures of the cardiac chambers (asterisk; ventricle). Legends: R: Right; L: Left; S: Superior side; I: Inferior side. Scale bars: 50 μm.

The online version of this article includes the following figure supplement(s) for figure 3:

**Figure supplement 1.** Transgenic *0.2Intr1spaw:eGFP* reporter line.

**Figure supplement 2.** Use of the *lft2BAC:Gal4FF* to track left- and -right originating cardiomyocytes during cardiac looping.

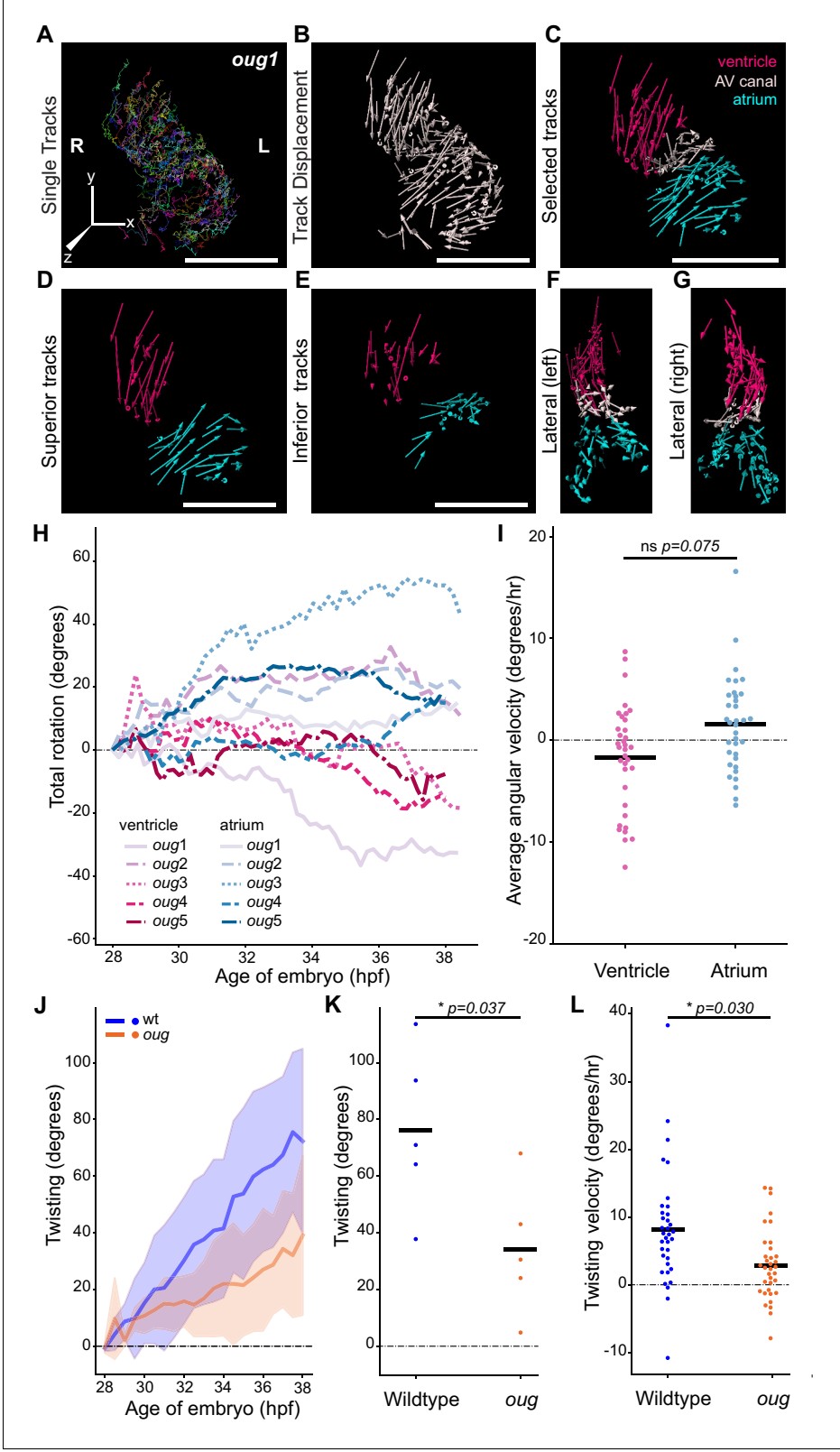

**Figure 4.** Cardiac looping is defective in *oug* mutants due to absence of asymmetric rotation of the cardiac chambers. (**A**) Total tracks (Ventral View) obtained from a time-lapse movie of cardiac looping in an *oug* mutant. Each track is colour-coded and is assigned an ID number. (**B**) Track displacement vectors for each trace drawn in (**A**). (**C**) Track displacement vectors to be analyzed are selected, categorized by visual inspection and colour-

*Figure 4 continued on next page*

*Figure 4 continued*

labeled accordingly. (**D**) Detail of the track displacement vectors on the superior cardiac side and (**E**) on the inferior cardiac side. (**F**), (**G**) Lateral views of the selected tracks reveal no major displacement along the Z-axis. (**H**) Cumulative rotation angle for the ventricle (shades of red) and atrium (shades of blue) in *oug* hearts. Compare with *Figure 2F*; the chambers do not show separation. With the outflow of the heart as viewpoint, positive values represent anti-clockwise rotation and negative values represent clockwise rotation. (**I**) Comparison of the average angular velocity for each replicate per 1.5 hr time window displayed by the chambers analyzed in (**H**). Horizontal bars: mean values. (**J**–**L**) Twisting of the heart tube during cardiac looping. (**J**) Plot of the twisting angle (as defined in the main text and in Appendix 1- Supplementary Methods) in time. The looping defect in *oug* is due to a reduced twisting of the heart tube. Solid lines: Mean; shaded area: standard deviation. (**K**) Average twisting angle for the sample hearts 9 hr after the start of the timelapse (37 hpf). Horizontal bars: mean values. (**L**) The twisting velocity in 1.5 hr windows in the wt samples is significantly higher than in *oug*. Horizontal bars: mean values. Scale bars: (**A–C**): 100 µm.

The online version of this article includes the following video, source data, and figure supplement(s) for figure 4:

**Source data 1.** Source files for data presented in panels H, I, J, K, and L.
**Figure supplement 1.** Analyzed *oug* hearts and AV canal analysis.
**Figure supplement 1—source data 1.** Source files for data presented in panel B of *Figure 4—figure supplement 1*.
**Figure 4—video 1.** Timelapse of *oug1 tg(myl7:Gal4FF; UAS:H2A-GFP)* heart.
https://elifesciences.org/articles/61733#fig4video1
**Figure 4—video 2.** Displacement vectors for *oug1* heart (360° rotation).
https://elifesciences.org/articles/61733#fig4video2
**Figure 4—video 3.** Computational processing of *oug1* timelapse.
https://elifesciences.org/articles/61733#fig4video3
**Figure 4—video 4.** Timelapse of *oug2 tg(myl7:Gal4FF; UAS:H2A-GFP)* heart.
https://elifesciences.org/articles/61733#fig4video4
**Figure 4—video 5.** Timelapse of *oug3 tg(myl7:Gal4FF; UAS:H2A-GFP)* heart.
https://elifesciences.org/articles/61733#fig4video5
**Figure 4—video 6.** Timelapse of *oug4 tg(myl7:Gal4FF; UAS:H2A-GFP)* heart.
https://elifesciences.org/articles/61733#fig4video6
**Figure 4—video 7.** Timelapse of *oug5 tg(myl7:Gal4FF; UAS:H2A-GFP)* heart.
https://elifesciences.org/articles/61733#fig4video7
**Figure 4—video 8.** Displacement vectors for *oug2* heart (360° rotation).
https://elifesciences.org/articles/61733#fig4video8
**Figure 4—video 9.** Displacement vectors for *oug3* heart (360° rotation).
https://elifesciences.org/articles/61733#fig4video9
**Figure 4—video 10.** Displacement vectors for *oug4* heart (360° rotation).
https://elifesciences.org/articles/61733#fig4video10
**Figure 4—video 11.** Displacement vectors for *oug5* heart (360° rotation).
https://elifesciences.org/articles/61733#fig4video11
**Figure 4—video 12.** Computational processing of *oug2* timelapse.
https://elifesciences.org/articles/61733#fig4video12
**Figure 4—video 13.** Computational processing of *oug3* timelapse.
https://elifesciences.org/articles/61733#fig4video13
**Figure 4—video 14.** Computational processing of *oug4* timelapse.
https://elifesciences.org/articles/61733#fig4video14
**Figure 4—video 15.** Computational processing of *oug5* timelapse.
https://elifesciences.org/articles/61733#fig4video15

difference (*Figure 4I*), with values for both chambers distributed in the positive and negative halves of the plot. These observations confirm that the strong reduction in reverse rotation of the chambers in *oug* embryos underlies the reduced cardiac looping. In fact, the values obtained for the chamber cardiomyocytes in *oug* are similar to those of the AV canal (compare *Figure 4—figure supplement 1* and *Figure 4H,I*), further supporting the lack of heart tube twisting in absence of *tbx5a*.

To assess the extent of the transformation in wild type and *oug* hearts, we calculated the twisting angle as the difference between rotation angles of the ventricle and the atrium from 28 to 38 hpf (*Figure 4J*, Supplementary *Equation 20*). Both the average twisting angle after 37 hpf (*Figure 4K*)

and twisting velocity throughout the time-lapse (*Figure 4L*) are significantly higher in wild type compared to *oug* hearts. From these results, we conclude that twisting of the chambers around the AV canal is a *tbx5a*-dependent process.

## A tissue intrinsic mechanism, and not cell addition to the embryonic cardiac poles, is required for torsion of the heart tube

Next, we asked which mechanisms could be driving the observed opposite twisting of the chamber around the AV canal during heart looping. During mouse heart morphogenesis, asymmetric contributions at the poles drive a helical rotation of the tube (*Le Garrec et al., 2017*). Although the zebrafish heart does not form a helix, we considered that the opposite chamber rotation could be driven by a similar mechanism. Previous work has demonstrated that also in zebrafish cells from the second heart field (SHF) are added to the poles of the heart tube concomitantly with cardiac looping (*de Pater et al., 2009*; *Lazic and Scott, 2011*; *Zhou et al., 2011*). To test whether cardiomyocyte addition from the SHF is required for the correct progression of cardiac looping, we abolished it in two independent manners prior to the onset of cardiac looping: (1) by treating embryos with the FGF inhibitor SU5402 (*de Pater et al., 2009*) and (2) by explanting linear heart tubes and culturing them ex vivo for 24 hr, as previously described (*Noël et al., 2013*). Treatment with SU5402 was efficient, as we counted reduced numbers of ventricular cardiomyocytes, confirming previous reports (*de Pater et al., 2009*; *Figure 5—figure supplement 1*). Cardiac looping was however not strongly affected, as SU5402-treated hearts displayed a clear S shape at 48 hpf, and left-originating cardiomyocytes could be observed at the outer curvature of the ventricle (*Figure 5A*). Moreover, quantification of the looping angle did not reveal any significant difference with the control condition (*Figure 5B*). In explanted cultured *tg(lft2BAC:Gal4FF; UAS:RFP; myl7:GFP)* hearts (*Figure 5C,D*), we also observed convincing cardiac looping (*Figure 5D*, upper panels). The use of the *lft2* reporter allowed us to orient the explanted heart tubes and observe that left-originating cardiomyocytes locate to the outer curvatures of the ventricle and atrium. We also exposed explanted heart tubes to SU5402 during culture and still observed satisfactory looping morphogenesis (*Figure 5D*, lower panels). From these observations, we concluded that heart tubes ex vivo not only retain their capacity to loop dextrally (*Noël et al., 2013*), but also that the cardiac torsion is still occurring.

Consistent with our observation that addition of SHF cells to the poles of the heart tube is dispensable for opposite chamber rotation and cardiac looping, we observed no changes in cardiomyocyte numbers in the ventricle (or atrium) of *oug* mutants (*Figure 6A,B*). To reject the possibility that the looping phenotype displayed by *oug* mutants is secondary to fluid pressure caused by the cardiac edema appearing by 2 dpf, we explanted *oug tg(myl7:Gal4FF; UAS:RFP; 0.2Intr1spaw:GFP)* heart tubes at 28 hpf. Indeed, after 24 hr in vitro culturing, *oug* mutant hearts failed to loop, indicating that the morphogenesis defect was not related to changes in physical properties of *oug* mutant embryos (*Figure 6C*). From the above results, we conclude that cardiomyocyte addition from the SHF is dispensable for cardiac looping.

## Reduced anisotropic growth in *oug* cardiomyocytes

Epithelial remodeling is an important driver for asymmetric rotation of the *Drosophila* gut tube or looping of the chick midgut and heart tube (*Taniguchi et al., 2011*; *Ray et al., 2018*; *Davis et al., 2008*). In the zebrafish heart tube changes in cardiomyocyte shape and cell boundaries occur during looping morphogenesis as well (*Merks et al., 2018*; *Auman et al., 2007*; *Lombardo et al., 2019*). Hence, we next proceeded by assessing the shape of GFP+ ventricular cardiomyocytes between 30 hpf and 42 hpf (*Figure 7A*; for wt: *Figure 7—videos 1–4*; for *oug*: *Figure 7—videos 5–7*). Indeed, we could determine that the progression of the left-originating cardiomyocytes is concomitant to anisotropic growth of these cardiomyocytes, which results in a reduced roundness (*Figure 7B*). Analysis of the positioning of cardiomyocytes at the border between left- (green) and right- (magenta) originating cardiac regions confirmed this change in cell shape (*Figure 7C–D*; for wt: *Figure 7—videos 8–11*; for *oug*: *Figure 7—videos 12–15*), possibly suggesting involvement of cell intercalation. In *oug* mutant embryos, we observed that ventricular cells retain their higher cell roundness throughout the analysis window and display a much straighter left/right boundary in the ventricle. We therefore conclude that our results are consistent with the proposed model in which tissue-intrinsic

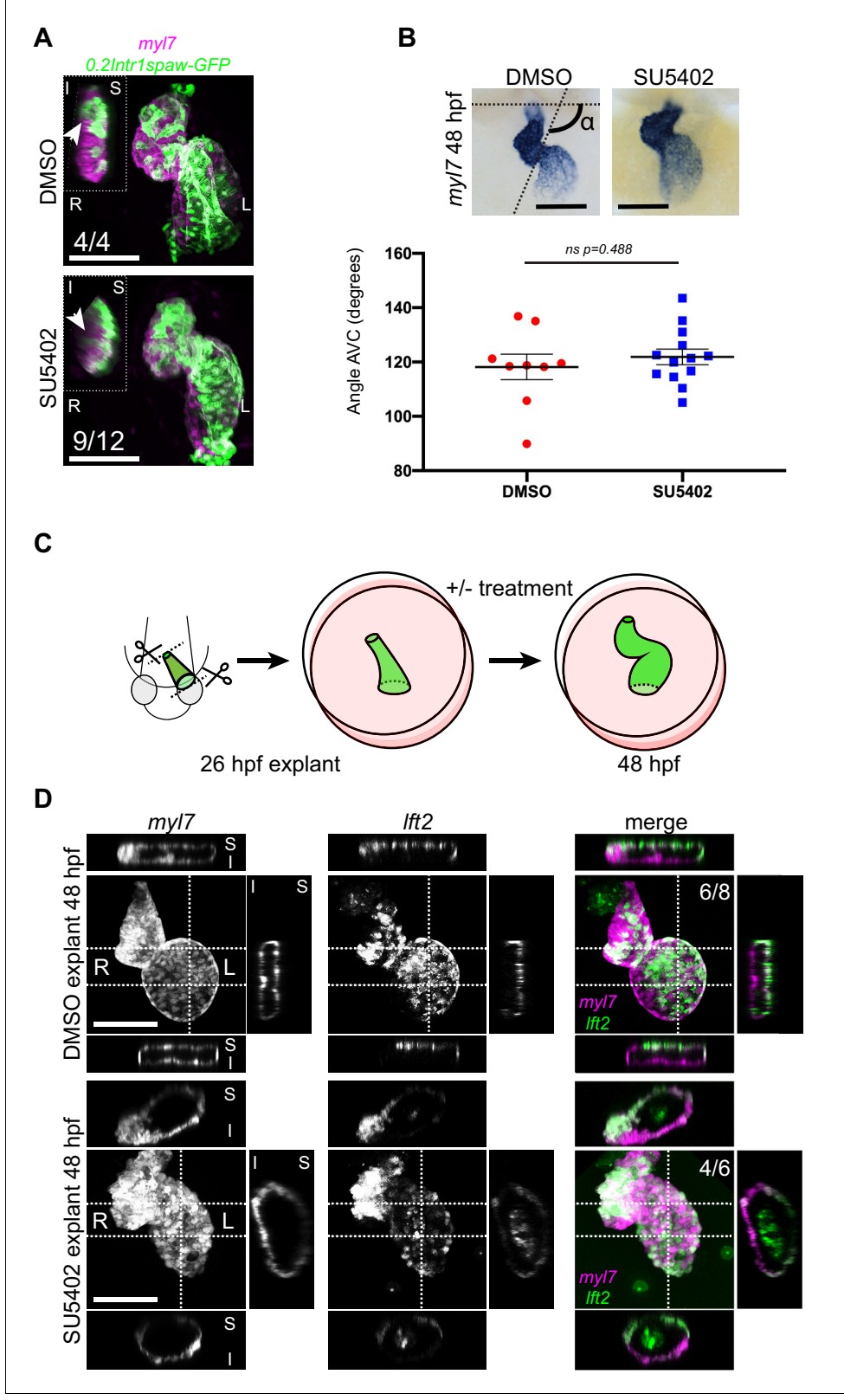

**Figure 5.** Chemical and physical suppression of cell addition to the heart tube do not affect proper completion of cardiac looping. Representative SU5402-treated and DMSO Control (explanted) hearts are shown. (**A**) 48 hpf *tg (myl7:Gal4FF; UAS:RFP; 0.2Intr1spaw-GFP)* hearts. In SU5402-treated hearts, dextral looping is completed and left-originating cardiomyocytes (green) can be observed at the ventricle outer curvature, similar to the control

*Figure 5 continued on next page*

*Figure 5 continued*

condition (arrowheads). (**B**) Quantification and comparison of AV canal angles in SU5402-treated and DMSO Control embryos. AV canal angle measurement is exemplified in the upper left panel. (**C**) Heart explant procedure: as cardiac jogging is completed (26 hpf) heart tubes are explanted and put into culture for approximately 24 hpf during which chemical treatments can be carried out. At 48 hpf, the hearts are imaged. (**D**) Heart tubes explanted at 26 hpf and subsequently cultured in liquid medium for 24 hr display normal formation of a ventricle, atrium and atrioventricular canal. The *lft2* reporter allows visualization of left-originating cells at the outer curvature of both ventricle and atrium, in control (DMSO) and treatment (SU5402) conditions. For (**B**) mean values ± SEM are shown. Legends: R: Right; L: Left; S: Superior side; I: Inferior side. Scale bars: 100 μm.

The online version of this article includes the following source data and figure supplement(s) for figure 5:

**Source data 1.** Source files for data presented in panel B.
**Figure supplement 1.** SU5402 treatment.

properties drive opposite chamber rotation and cardiac looping (*Noël et al., 2013*; *Merks et al.,*

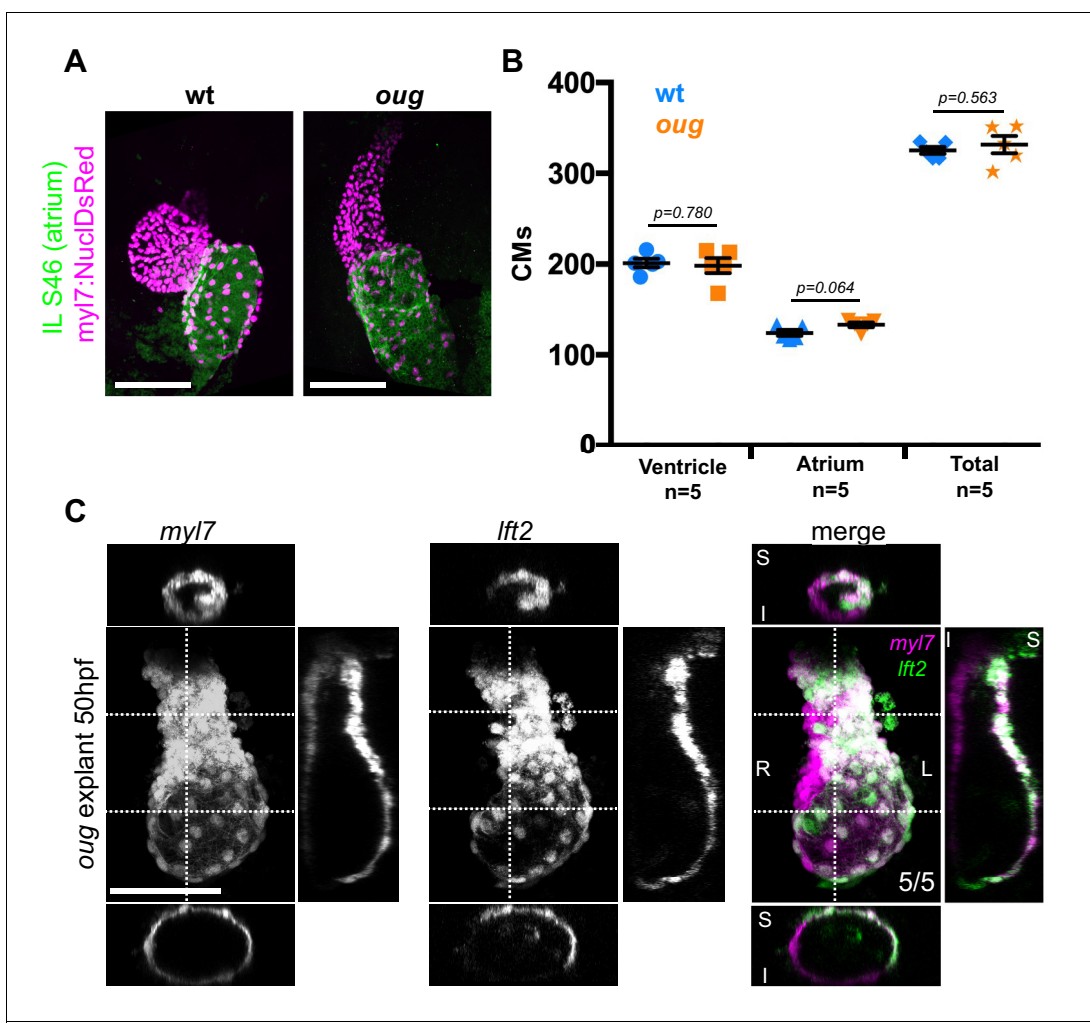

**Figure 6.** Defective looping in *oug mutants* is not due to reduced cardiomyocyte number or embryonic environment. (**A**) Immunofluorescence with atrium-specific S46 antibody allows distinction of the cardiac chambers. (**B**) Quantification of ventricular and atrial cardiomyocytes in wt and *oug* mutant embryos at 2dpf. (**C**) Explanting *oug* mutant hearts and culturing them in vitro, ex-embryo does not rescue defective looping. (**B**): Horizontal bars: mean value ± SEM. Legends: R: Right; L: Left; S: Superior side; I: Inferior side . Scale bars: 100 μm.

The online version of this article includes the following source data for figure 6:

**Source data 1.** Source files for data presented in panel B.

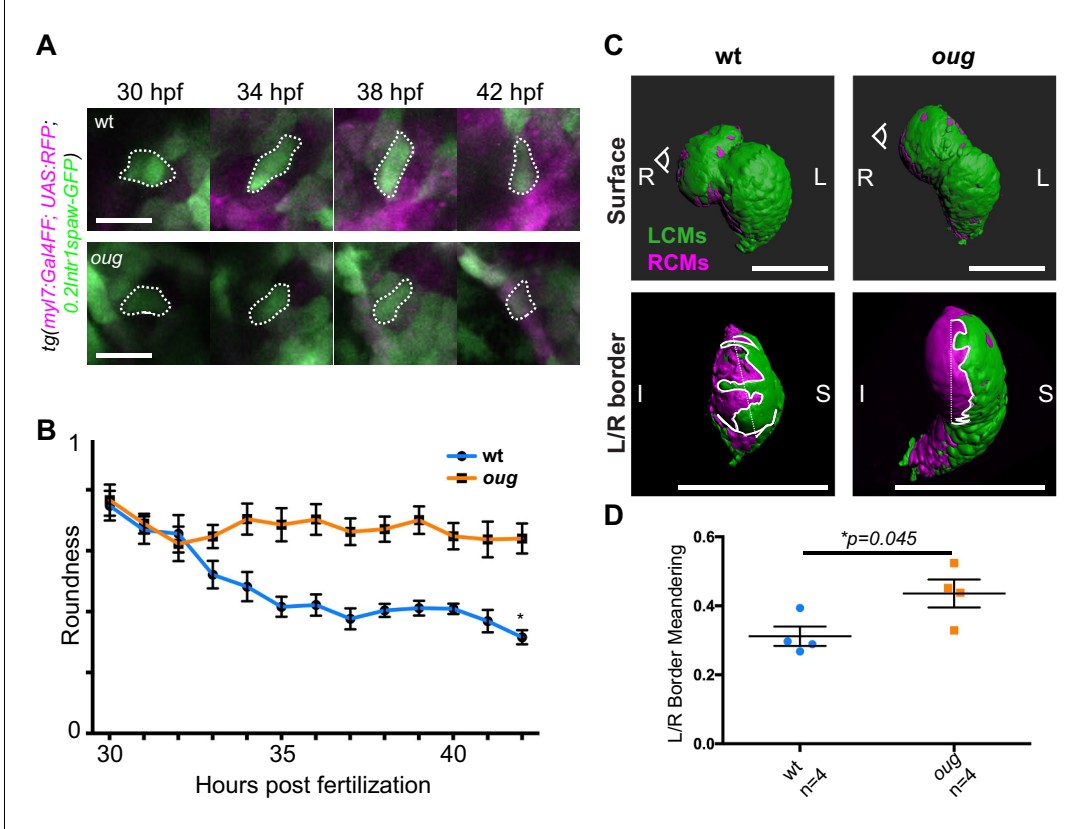

**Figure 7.** Anisotropic cell shape changes accompany cardiac looping. (**A**) Outline of ventricular cardiomyocytes assessed for assessed for cell roundness. Representative images of the data quantified in (**B**) are shown for wt (upper row) and *oug* (lower row). (**B**) Quantification of cell roundness as observed in (**A**) and comparison between values for wt and *oug* mutants. (**C**) Upper panels: surface rendering of *tg(myl7:Gal4FF; UAS:RFP; 0.2Intr1spaw-GFP)* in 48 hpf hearts allows clear definition of a boundary between Left-originating cardiomyocytes (LCMs, green) and right-originating cardiomyocytes (RCMs, magenta). This allows calculation of the straightness index of the left/right boundary (white) of the ventricle (lower panels, respective viewpoint indicated in upper panels). The straightness index is calculated as the ratio between distance between start and end point of left/right boundary at (straight dotted line) and length of left/right boundary measured on the ventricular surface. (**D**) Quantification of the straightness index is indicative of the level of anisotropic growth in wt and *oug* mutant hearts. (**B**) and (**D**): Horizontal bars: mean value ± SEM. Legends: R: Right; L: Left; S; Superior side; I: Inferior side. Scale bars: (**A**) 20 μm; (**C**) 100 μm.

The online version of this article includes the following video and source data for figure 7:

**Source data 1.** Source files for data presented in panels B and D.

**Figure 7—video 1.** Timelapse of wt *tg(myl7:Gal4FF; UAS:RFP; 0.2Intr1spaw-GFP)* heart 1 used for cardiomyocyte roundness calculations (*Figure 7A,B*).
https://elifesciences.org/articles/61733#fig7video1

**Figure 7—video 2.** Timelapse of wt *tg(myl7:Gal4FF; UAS:RFP; 0.2Intr1spaw-GFP)* heart 2 used for cardiomyocyte roundness calculations (*Figure 7A,B*).
https://elifesciences.org/articles/61733#fig7video2

**Figure 7—video 3.** Timelapse of wt *tg(myl7:Gal4FF; UAS:RFP; 0.2Intr1spaw-GFP)* heart 3 used for cardiomyocyte roundness calculations (*Figure 7A,B*).
https://elifesciences.org/articles/61733#fig7video3

**Figure 7—video 4.** Timelapse of wt *tg(myl7:Gal4FF; UAS:RFP; 0.2Intr1spaw-GFP)* heart 4 used for cardiomyocyte roundness calculations (*Figure 7A,B*).
https://elifesciences.org/articles/61733#fig7video4

**Figure 7—video 5.** Timelapse of *oug tg(myl7:Gal4FF; UAS:RFP; 0.2Intr1spaw-GFP)* heart 1 used for cardiomyocyte roundness calculations (*Figure 7A,B*).
https://elifesciences.org/articles/61733#fig7video5

**Figure 7—video 6.** Timelapse of *oug tg(myl7:Gal4FF; UAS:RFP; 0.2Intr1spaw-GFP)* heart 2 used for cardiomyocyte roundness calculations (*Figure 7A,B*).
https://elifesciences.org/articles/61733#fig7video6

**Figure 7—video 7.** Timelapse of *oug tg(myl7:Gal4FF; UAS:RFP; 0.2Intr1spaw-GFP)* heart 3 used for cardiomyocyte roundness calculations (*Figure 7A,B*).
https://elifesciences.org/articles/61733#fig7video7

**Figure 7—video 8.** 360° rotation animation of surface-rendered wt *tg(myl7:Gal4FF; UAS:RFP; 0.2Intr1spaw-GFP)* heart analyzed in *Figure 7C,D*.
https://elifesciences.org/articles/61733#fig7video8

**Figure 7—video 9.** 360° rotation animation of surface-rendered wt *tg(myl7:Gal4FF; UAS:RFP; 0.2Intr1spaw-GFP)* heart analyzed in *Figure 7C,D*.
https://elifesciences.org/articles/61733#fig7video9

*Figure 7 continued on next page*

*Figure 7 continued*

**Figure 7—video 10.** 360° rotation animation of surface-rendered wt *tg(myl7:Gal4FF; UAS:RFP; 0.2Intr1spaw-GFP)* heart analyzed in *Figure 7C,D*.
https://elifesciences.org/articles/61733#fig7video10

**Figure 7—video 11.** 360° rotation animation of surface-rendered wt *tg(myl7:Gal4FF; UAS:RFP; 0.2Intr1spaw-GFP)* heart analyzed in *Figure 7C,D*.
https://elifesciences.org/articles/61733#fig7video11

**Figure 7—video 12.** 360° rotation animation of surface-rendered *oug tg(myl7:Gal4FF; UAS:RFP; 0.2Intr1spaw-GFP)* hearts analyzed in *Figure 7C,D*.
https://elifesciences.org/articles/61733#fig7video12

**Figure 7—video 13.** 360° rotation animation of surface-rendered *oug tg(myl7:Gal4FF; UAS:RFP; 0.2Intr1spaw-GFP)* hearts analyzed in *Figure 7C,D*.
https://elifesciences.org/articles/61733#fig7video13

**Figure 7—video 14.** 360° rotation animation of surface-rendered *oug tg(myl7:Gal4FF; UAS:RFP; 0.2Intr1spaw-GFP)* hearts analyzed in *Figure 7C,D*.
https://elifesciences.org/articles/61733#fig7video14

**Figure 7—video 15.** 360° rotation animation of surface-rendered *oug tg(myl7:Gal4FF; UAS:RFP; 0.2Intr1spaw-GFP)* hearts analyzed in *Figure 7C,D*.
https://elifesciences.org/articles/61733#fig7video15

*2018*) and indicate that Tbx5a activity is required for this to occur.

## Cardiac looping is reestablished in Tbx5a-defective hearts by suppression of Tbx2b activity

AV canal versus chamber specification is tightly regulated by a balance in gene activation and repression by Tbx5 and Tbx2, respectively (*Chi et al., 2008*; *Christoffels et al., 2004a*, reviewed in *Greulich et al., 2011*). As we observed an expansion of *tbx2b* expression in *oug* mutant hearts (*Figure 1D*), we first tested whether the myocardial patterning defect in *oug* mutants could be rescued by reducing Tbx2b activity. To do so, we used the *tbx2b* mutant *from beyond* (*fby*) (*Snelson et al., 2008*). Analysis of cardiac markers by ISH and transgenic reporters revealed that *fby/tbx2b⁻/⁻* embryos display robust cardiac looping and a properly patterned heart (*Figure 8* and *Figure 8—figure supplement 1*). In *tbx5a⁻/⁻;tbx2b⁻/⁻* (*oug/fby*) double mutant background, ISH indicated rescue of the constriction at the AV canal (*Figure 8A*), reestablishment of *nppa* expression in the cardiac chambers, while *bmp4* expression remained similar to that of *tbx5a-/-* hearts (*Figure 8—figure supplement 1*). Analysis of *tg(nppaBAC:mCitrine)* in vivo confirmed the rescue of *nppa* expression in the atrium of *tbx5a⁻/⁻;tbx2b⁻/⁻* double mutants, which was absent in *oug* embryos (*Figure 8B*). Next, we investigated how the rescue in cardiac patterning affects heart looping morphogenesis. Along with the reestablishment of myocardial patterning, we also observed a significant rescue of the looping phenotype by measuring the looping angle (*Figure 8C*). Consistently with these observations, analysis of *tg(myl7:Gal4FF; UAS:RFP; 0.2Intr1spaw-GFP)* in *tbx5a⁻/⁻;tbx2b⁻/⁻* embryonic hearts revealed the presence of GFP+ left-originating cardiomyocytes on the inferior side of the ventricle (*Figure 8D–D′′′*), indicating substantial rescue of the twisting of the heart tube. Additionally, we observed that while pectoral fin development was not rescued in *tbx5a⁻/⁻;tbx2b⁻/⁻* double mutants, these fish hardly developed a cardiac edema, as compared to *oug* mutants (*Figure 8—figure supplement 2*). Altogether, these results indicate that heart looping morphogenesis is the result of proper tissue patterning and requires a finely balanced Tbx5a and Tbx2b activity.

## Discussion

In this study, we have analyzed the early phase of cardiac looping, from its onset at the end of cardiac jogging (28 hpf) until approximately 40 hpf, as the heart tube acquires a distinct S-shape. As knowledge about the cardiomyocyte behavior during these initial stages of heart looping was limited, we carried out a detailed and quantitative four-dimensional analysis of cellular trajectories in the different heart segments, in order to better understand how these underlie the looping transformation at the organ level. By calculating the angular velocity of ventricular and atrial cardiomyocytes, we establish that the two chambers rotate in opposing directions with respect to their longitudinal axes (*Figure 2*), essentially twisting around the AV canal region. When this twisting of the heart tube is defective, as in *oug/tbx5a* (*Figure 4*), cardiac looping is reduced or absent. Combination of these results with the genetic tracing of left-originating cardiomyocytes allowed us to formulate a model for cardiac looping in the zebrafish (*Figure 9*). Finally, we conclude that twisting of the heart tube is

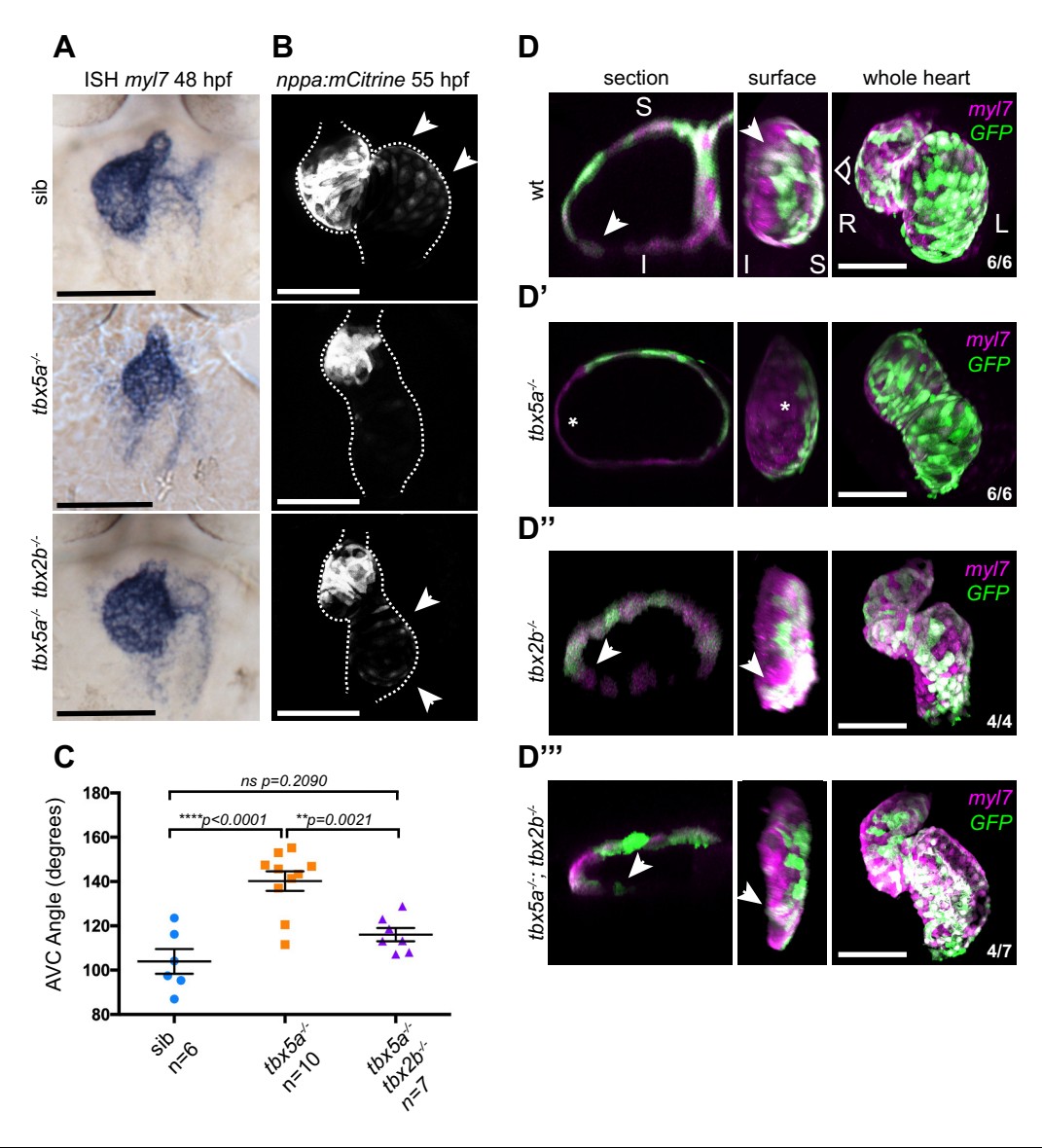

**Figure 8.** Defective cardiac looping in *oug* mutants is alleviated by simultaneous loss of *tbx2b*. (**A**) ISH for *myl7* at 50 hpf in wild type siblings, *oug* mutants and *tbx5a;tbx2b* double mutants. (**B**) Confocal maximum projections of 2dpf *tg(nppa:mCitrine)* hearts. In the *tbx5a;tbx2b* double mutants, atrial expression of *nppa*, which was lost in *oug* mutants, is re-instated. (**C**) Quantification and comparison of AV canal angles in wild-type siblings, *tbx5a* mutants and *tbx5a;tbx2b* double mutants. Quantification of AV canal angle is carried out as reported in *Figure 5D*. (**D–D'''**) 48 hpf *tg(myl7:Gal4FF; UAS:RFP; 0.2Intr1spaw-GFP)* hearts. Wt (**D**) and *tbx5⁻/⁻* (**D'**) are shown for comparison. *tbx2b⁻/⁻* hearts (**D''**) display robust dextral looping and left-originating cardiomyocytes (green) at the ventricle outer curvature, similar to wt (arrowheads in D; *Figure 3B*). In double homozygous mutants *tbx5a⁻/⁻; tbx2b⁻/⁻* (**D'''**) rescue of cardiac looping is observed, accompanied by presence of left-originating cardiomyocytes at the ventricle OC (Compare with D, **D''**). (**C**): Horizontal bars: mean value ± SEM. Legends: R: Right; L: Left; S: Superior side; I: Inferior side. Scale bars: 100 μm.

The online version of this article includes the following source data and figure supplement(s) for figure 8:

**Source data 1.** Source files for data presented in panel C.
**Figure supplement 1.** Analysis of cardiac markers in *fby/tbx2b⁻/⁻ and tbx5a⁻/⁻; tbx2b⁻/⁻* embryos at 50 hpf.
**Figure supplement 2.** Phenotypical analysis of *tbx5a⁻/⁻; tbx2b⁻/⁻* larvae at 6 dpf.

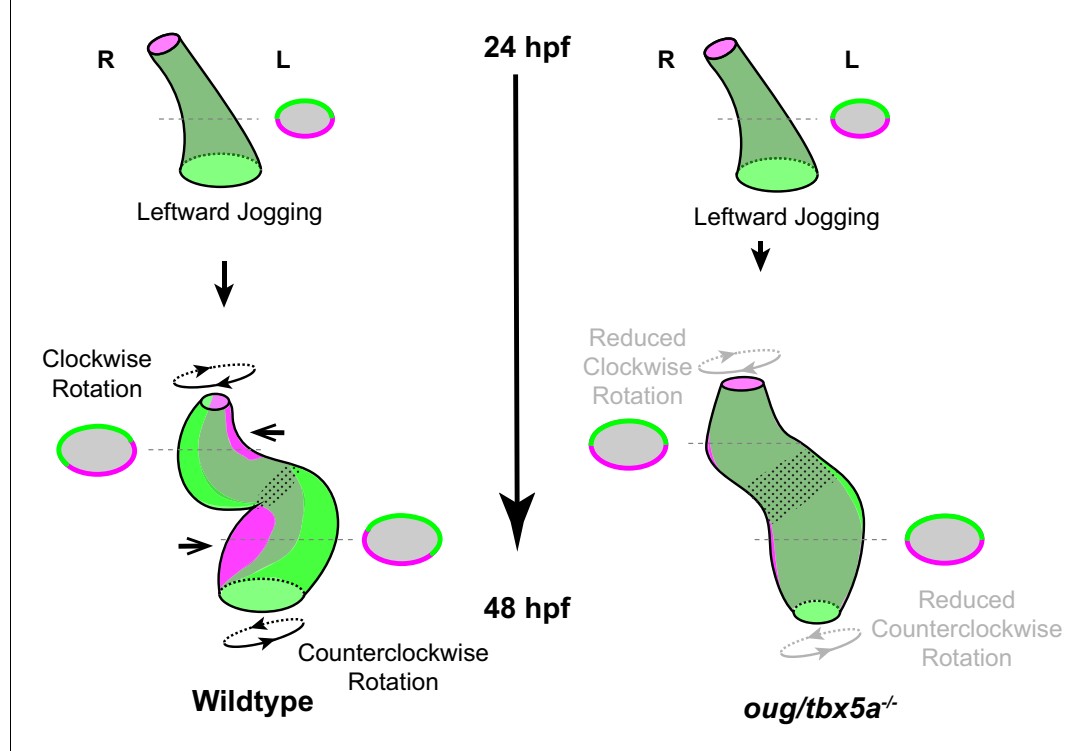

**Figure 9.** Model for cardiac looping morphogenesis. Viewpoint for describing direction of rotation is always the outflow tract (OFT). Left- and right-originating regions of the embryonic myocardium are reported in green and magenta, respectively. Transversal sections are shown next to the corresponding cartoon. In wild-type hearts, at the end of cardiac jogging, twisting of the heart tube results in disposition of left-originating cardiomyocytes toward the outer curvatures of both the ventricle and atrium. The resulting twisting of the heart tube is driven by the clockwise rotation of the ventricle and counterclockwise rotation of the atrium, around a fixed hinge, the AV canal. In *oug* hearts, cardiac jogging is completed properly, but progression of cardiac looping is defective. Reduced twisting of the heart tube and chamber expansion are observed. Defective looping is accompanied by an expansion of the expression domain of *tbx2b* (spotted pattern), especially noticeable at the AV canal (see also *Figure 1H*). Legends: R: Right; L: Left.

a tissue intrinsic process that requires proper patterning into chamber and AV canal myocardium, which is regulated by T-box containing transcription factors.

In this study, we identified a novel *tbx5a* allele, *oug*, which we demonstrated to be a *tbx5a* null allele (*Figure 1J*). Indeed, in *oug* approximately 75% of the gene product is lost, including a large portion of the DNA-binding T-box domain. In *oug* mutants, we observed an expansion of genes that mark the AV canal (*Figure 1H*). Work in various vertebrate models has established that Tbx5 has a crucial role in cardiomyocyte differentiation and establishment of the working chamber (*Steimle and Moskowitz, 2017* and references therein). In mouse, this role is balanced by other T-box factors, such as Tbx2/3 (*Habets et al., 2002*; *Hoogaars et al., 2007a*; *Hoogaars et al., 2007b*), which compete for the same T-box sequences as Tbx5 and are restricted to non-chamber myocardium (i.e. AV canal) (*Shirai et al., 2009*). In the zebrafish *oug* mutant, the absence of Tbx5a results in the expansion of the AV canal as illustrated by expanded domains of expression of *tbx2b*, *bmp4*, and *has2*, as is also observed in other zebrafish looping mutants (*Hurlstone et al., 2003*; *Smith et al., 2011b*). In *hst* mutants, however, the picture seems less clear (*Figure 1H*). Based on the *hst* results, a model was proposed in which Tbx5a stimulates the expression of *tbx2* in the AV canal (*Garrity et al., 2002*; *Camarata et al., 2010*), which needs to be reconsidered based on the *oug* results presented here. These different outcomes in patterning of the AV canal and chamber myocardium might be explained by the different locations of the *oug* and *hst* mutations in *tbx5a* (*Figure 1F*). While in *oug/tbx5a* the T-box is truncated, it is still present in *hst/tbx5a* (*Garrity et al., 2002*), which is only missing regions proposed to affect its subcellular localization (*Camarata et al., 2010*).

There is a striking resemblance between the rotation in the ventricle during looping as described here and the clockwise rotation that occurs earlier when the cardiac disc transforms into a linear

heart tube, which has been described in several studies (*Baker et al., 2008*; *Smith et al., 2008*; *de Campos-Baptista et al., 2008*). As a consequence of this first rotation event, the original left-right orientation of the cardiac cells is transformed to a superior-inferior orientation. In a previously published study, the authors suggested that after the linear heart tube is formed this superior-inferior orientation is transformed back to the original left-right orientation due to a second counter-clockwise rotation around its longitudinal axis (*Baker et al., 2008*). Although we detected atrial cardiomyocyte movement compatible with this observation (*Figure 2*), we did not observe this second rotation when tracing the ventricular cardiomyocytes originating from the left and right lateral plate mesoderm. This difference between the observations might be partially explained by how the left and right cardiac cells were labeled in the two studies. In our study, we used stable transgenic lines in which *lefty2* or *spaw* regulatory elements drive left-sided expression of GFP. In the original study by *Baker et al., 2008*, a *myl7:Dendra* plasmid was injected at the one- or two-cell stage and embryos were screened before 18 hpf for either left- or right sided expression and analysed at 48 hpf. As we know now, at 18 hpf, the *myl7* promoter is only activated in the first heart field (FHF). Cardiomyocytes from the second heart field (SHF) initiate *myl7* expression at a later stage, up to 38 hpf, when these are added to the cardiac poles (*de Pater et al., 2009*; *Lazic and Scott, 2011*). As a consequence, embryos scored with unilateral *myl7:Dendra* expression at 18 hpf may display expression of Dendra in cardiomyocytes from the originally (18 hpf) non-expressing side when scored at 48 hpf. The gradual activation of *myl7* due to the continuous process of cardiomyocyte differentiation during heart tube morphogenesis limits its use as a cell tracing technique.

The clockwise rotation we observed in the ventricle is in the same direction as the rotation that was observed during linear heart tube formation (*Smith et al., 2008*). Recently, a clockwise rotation was also described in the OFT of the zebrafish heart at later cardiac looping stages (40–54 hpf) (*Lombardo et al., 2019*). Together, these observations suggest that a clockwise rotation of the cardiac tissue is initiated during linear heart tube formation (20–26 hpf) and that this clockwise rotation continues in the ventricle (28–42 hpf) during looping initiation and continues in the OFT (40–54 hpf) during the late looping stage. In the atrium, however, we describe here a counterclockwise rotation during the early looping phase (28–42 hpf), resulting in a torsion of the heart tube.

During cardiac looping, there is extensive growth of the myocardium. Due to the addition of cells at the poles from the SHF, the number of cardiomyocytes is doubled between 24 and 48 hpf (*de Pater et al., 2009*). Reduced cell addition from the SHF by inhibiting FGF signaling still allowed looping and twisting of the zebrafish heart tube (*Figure 5*). This is different in the mouse heart, where reduced growth due to compromised addition of cells from the SHF results in looping defects (*Cai et al., 2003*; *Cohen et al., 2012*; *Tsuchihashi et al., 2011*). This may be due to more extensive growth of the murine heart, which extends its length over fourfold during looping, resulting in a distinct helical shape (*Le Garrec et al., 2017*).

Our data builds upon previous work exploring the intrinsic capacity of the heart to loop (*Noël et al., 2013*; *Ray et al., 2018*; *Honda et al., 2020*). Corroborating such a model, we observed that the twisting and looping of the heart tube still occurs in explanted hearts, or if SHF contribution is chemically inhibited. We therefore conclude that the early phase of heart looping in zebrafish occurs independently of cell addition. Other examples of tubes that undergo looping morphogenesis due to intrinsic LR asymmetry are the *Drosophila* genitalia and hindgut (*Sato et al., 2015*; *Taniguchi et al., 2011*). For these tubes, it is proposed that intrinsic chirality of the cells drive looping morphogenesis. In the zebrafish, the outer layer of the heart tube, the myocardium, is organized with distinct apical-basal polarity (*Bakkers et al., 2009*). During heart looping and chamber ballooning, the myocardium undergoes remodeling, which coincides with regional cell shape changes (*Merks et al., 2018*; *Auman et al., 2007*; *Lombardo et al., 2019*). Interestingly, defective chamber expansion is accompanied in *oug* embryos by failure of the cardiomyocytes of the ventricle to remodel anisotropically, a process that is regulated by non-canonical Wnt-and PCP-signaling (*Merks et al., 2018*). Although regulation by Tbx5 of canonical Wnt ligands is established in limb (*Takeuchi et al., 2003*; *Ng et al., 2002*) and lung (*Steimle et al., 2018*) development, a potential role in controlling cardiac non-canonical Wnt signaling still needs to be explored.

In *oug* mutants, *nppa* expression was reduced while *tbx2b* expression was expanded in the AV canal. This was restored in in *tbx5a*$^{-/-}$;*tbx2b*$^{-/-}$ (*oug/fby*) double mutants, which is consistent with the proposed roles of Tbx5 and Tbx2 in patterning the heart in chamber myocardium and primary (e.g. AV canal) myocardium (*Christoffels et al., 2004b*). In this respect, it is surprising that no cardiac

phenotype was observed in *fby/tbx2b* mutants (*Figure 8*; *Figure 8—figure supplement 1*). This could be ascribed to the presence in zebrafish of a second *tbx2* paralogue, *tbx2a*, which is also expressed in the embryonic heart (*Ribeiro et al., 2007*). The observed looping defects in *oug* in combination with the observed rescue of cardiac looping in *oug/fby* double mutant supports a model in which cardiac patterning in chamber and AV canal myocardium is an important driver for the intrinsic heart looping morphogenesis.

# Materials and methods

## Key resources table

| Reagent type (species) or resource | Designation | Source or reference | Identifiers | Additional information |
|---|---|---|---|---|
| Gene (*Danio rerio*) | *tbx5a* | NA | ZDB-GENE-991124–7 | |
| Strain, strain background (*Danio rerio*) | Tübingen Long Fin (TL) | ZIRC | ZDB-GENO-990623–2 | |
| Genetic reagent (*Danio rerio*) | *oug/tbx5a* | This paper | | More info on generation of this line can be found in the Materials and Methods section. |
| Genetic reagent (*Danio rerio*) | *hst/tbx5a* | ZIRC | ZDB-ALT-030627–2 | |
| Genetic reagent (*Danio rerio*) | *fby/tbx2b* | ZIRC | ZDB-ALT-070117–1 | |
| Genetic reagent (*Danio rerio*) | *tg(myl7:Gal4FF)* | DOI: 10.1242/dev.113894 | ZDB-ALT-151008–1 | |
| Genetic reagent (*Danio rerio*) | *tg(lft2BAC:Gal4FF)* | DOI: 10.1093/cvr/cvab004 | Not available | |
| Genetic reagent (*Danio rerio*) | *tg(UAS:RFP)* | DOI: 10.1073/pnas.0704963105 | ZDB-ALT-080528–2 | |
| Genetic reagent (*Danio rerio*) | *tg(UAS:H2A-GFP)* | DOI: 10.1242/dev.113894 | ZDB-ALT-151008–2 | |
| Genetic reagent (*Danio rerio*) | *tg(myl7:dsRed)*^s879Tg^ | DOI: 10.1101/gad.1629408 | ZDB-FISH-150901–3078 | |
| Genetic reagent (*Danio rerio*) | *tg(mCitrine:nppa)* | DOI: 10.7554/eLife.50163 | ZDB-ALT-201116–10 | |
| Cell line (*Chlorocebus aethiops*) | kidney fibroblast-like cell line (SV 40 transformed, Adult) | ATCC | Cat# CRL-1651; RRID: CVCL_0224 | |
| Transfected construct (*Chlorocebus aethiops*) | pGL3-Basic (plasmid) | Promega | Cat# E1751; Genbank: U47295 | |
| Transfected construct (*Chlorocebus aethiops*) | phRG-TK Renilla (plasmid) | Promega | Cat# E6291; Genbank: AF362551 | |
| Antibody | Living Colors anti-DsRed (Rabbit polyclonal) | Takara Bio | Cat# 101004; RRID:AB_10013483 | 1:200 |
| Antibody | Myosin heavy chain, slow developmental (Mouse monoclonal) | DSHB | Cat# s46, RRID:AB_528376 | 1:200 |
| Antibody | Anti-GFP (Chicken polyclonal) | Aves Labs | Cat# GFP-1010, RRID:AB_2307313 | 1:500 |
| Antibody | Anti-Digoxigenin-AP, Fab fragments (Sheep polyclonal) | Roche | Cat# 11093274910, RRID: AB_2734716 | 1:5000 |
| Antibody | Anti-Fluorescein-AP, Fab fragments (Sheep polyclonal) | Roche | Cat# 11426338910, RRID: AB_2734723 | 1:5000 |

*Continued on next page*

*Continued*

| Reagent type (species) or resource | Designation | Source or reference | Identifiers | Additional information |
|---|---|---|---|---|
| Recombinant DNA reagent | E1b-GFP-Tol2-Gateway | DOI: 10.1101/gr.133546.111 Obtained from Addgene | RRID:Addgene_37846 | |
| Sequence-based reagent | Start site morpholino: *tnnt2a* | DOI: 10.1038/ng875 | ZDB-MRPHLNO-060317–4 | 5' - CATGTTTGCTCTGA TCTGACACGCA - 3' 2 ng / embryo |
| Commercial assay or kit | NBT/BCIP Stock solution | Sigma-Aldrich | Cat# 11681451001 | |
| Commercial assay or kit | INT/BCIP Stock solution | Sigma-Aldrich | Cat# 11681460001 | |
| Chemical compound, drug | SU5402 | Sigma-Aldrich | Cat# 572630; CAS 215543-92-3 | 10 µM |
| Chemical compound, drug | phenylthourea | Sigma-Aldrich | Cat# P7629; CAS103-85-5 | 0,003%(v/v) |
| Software, algorithm | Fiji | https://fiji.sc/ | RRID:SCR_002285 | |
| Software, algorithm | Volocity 3D Image Analysis Software | Perkin Elmer | RRID:SCR_002668 | |
| Software, algorithm | Graphpad Prism 9.0 | Graphpad | RRID:SCR_002798 | V9.0 |
| Software, algorithm | Imaris data visualization software | Bitplane | RRID:SCR_007370 | V9.3.1 |
| Software, algorithm | heartbending.py | Source or reference: custom software, available in public repository: https://github.com/rmerks/heartbending (copy archived at swh:1:rev:149f05441e06f875faa3f9ab21101619bce25e93; *Tsingos, 2021*) | commit 149f054 | Code for transforming cell track data and for statistical analysis of cell rotation around the heart segment axes. |

## Zebrafish lines

All animal experiments were conducted under the guidelines of the animal welfare committee of the Royal Netherlands Academy of Arts and Sciences (KNAW). Adult zebrafish (*Danio rerio*) were maintained and embryos raised and staged as previously described (*Aleström et al., 2020*; *Westerfield, 1993*).

The zebrafish lines used in this study are Tübingen longfin (wild type), *hst/tbx5a* (*Garrity et al., 2002*), *fby/tbx2b* (*Snelson et al., 2008*), *tg(myl7:Gal4FF)* (*Strate et al., 2015*); *tg(lft2BAC:Gal4FF)* (*Derrick et al., 2021*); *tg(UAS:RFP)* (*Asakawa et al., 2008*); *tg(UAS:H2A-GFP)* (*Strate et al., 2015*); *tg(myl7:DsRed)* (*Mably et al., 2003*); *tg(mCitrine:nppa)* (*Honkoop et al., 2019*).

## Positional cloning of *oudegracht/tbx5a*

The *oudegracht/tbx5a^hu6499^* allele was identified in a ENU mutagenesis screen performed as described in *Wienholds et al., 2003*. The *oudegracht/tbx5a^hu6499^* was mapped using standard simple sequence length polymorphisms (SSLPs)-based meiotic mapping with SSLP primer sequences as pictured in *Figure 4*. The *oudegracht/tbx5a^hu6499^* mutation introduces a G to A substitution in Exon 4 of *tbx5a* (ENSDARG00000024894) resulting in the introduction of a premature stop codon. The mutation is identified by PCR amplification from genomic DNA using primers FKK106: 5'-GCGCA TCAGGTCTGTGAC-3' and FKK108: 5'-CCAAATACAAGTCCTCAAAGTG-3' followed by BtscI restriction of the PCR product. The oudegracht/tbx5a^hu6499^ mutation removes a BtscI restriction site.

## Generation of the *tg(0.2Intr1spaw:GFP)* transgenic line

A 228 bp conserved sequence located in intron 1 of *spaw* (ENSDARG00000014309) was amplified by PCR using primers FT294 5'-AGTCAAGCATCTCGGGAAGA-3' and FT295 5'-AGGTCCTGTCA-GAGCAGATG-3'. The resulting PCR product was subsequently cloned in the E1b-GFP-Tol2-Gateway

construct (Addgene #37846; *Birnbaum et al., 2012*) by Gateway cloning. The resulting construct was co-injected with 25 ng/µl Tol2 RNA in 1 cell zebrafish TL embryos. Founder fish (F0) were identified by outcrossing and the progeny (F1) was grown to establish the transgenic line.

## Microinjection of antisense morpholino

The *tnnt2a* morpholino oligonucleotide targeting the translation start site (5' - CATGTTTGCTCTGA TCTGACACGCA - 3') was used to block heart beat (*Sehnert et al., 2002*). We injected approximately 2 ng of the oligo morpholino in one-cell stage embryos.

## Chemical treatments

### SU5402 treatment

Embryos were dechorionated and treated with SU5402 (Sigma-Aldrich) at a concentration of 10 µM in E3 embryo medium from 24 hpf until 48 hpf at 28.5˚C. Control embryos were treated with the corresponding DMSO concentration.

## Phenylthiourea

Addition of phenylthiourea (PTU) at a concentration of 0.003% (v/v) to the E3 embryonic medium after shield stage (8 hpf) blocked pigmentation for improved confocal analysis.

## Heart explants

Zebrafish heart tubes were manually dissected from 26 hpf embryos using forceps and placed into supplemented L15 culture medium (Gibco-BRL; 15% fetal bovine serum, 0.8 mM $CaCl_2$, 50 µg/ml penicillin, 0.05 mg/ml streptomycin, 0.05 mg/ml gentomycin) essentially as described in *Noël et al., 2013*. Explants were incubated at 28.5˚C for 24 hr and fixed in 4% PFA overnight. Chemical treatment of the explants was carried out in an identical way as for the embryos. Explanted hearts were mounted in Vectashield (Vector Laboratories) before imaging.

## Immunofluorescent labeling

Zebrafish embryos at the appropriate developmental stage were fixed overnight in 2% paraformaldehyde (PFA) in PBS at 4˚C. After washing with 1 × PBS–Triton X-100 (0.1%; PBS-T) and blocking in 10% goat serum in 1 × PBST (blocking buffer;BB), embryos were incubated overnight at 4˚C with rabbit anti-DsRed (1:500 in BB; Takara Bio 632496), mouse anti-Myh6 antibody (1:200 in BB, DSHB, S46), or chicken anti-GFP (1:500 in BB, Aves Labs, GFP-1010). After washing in PBST, the embryos were incubated overnight at 4˚C in Cy3-conjugated goat anti-rabbit antibody (1:500 in BB; Jackson Immunoresearch, 111-165-144), Alexa488-conjugated goat anti-mouse (1:500 in BB, Invitrogen, A21133) or Alexa488-conjugated goat-anti-chicken (1:500 in BB; Invitrogen, A11039). Embryos were washed in PBST before imaging.

## Whole mount mRNA in situ hybridization (ISH)

Fixation of the embryos was carried overnight in 4% paraformaldehyde (PFA). Embryos were subsequently stored in methanol (MeOH) at −20˚C. Rehydration was carried out in PBST (PBS plus 0.1% Tween-20) and, depending on the stage, embryos were treated with 1 µg ml-1 Proteinase K (Promega) between 1 and 20 min. Embryos were then rinsed in PBST, post-fixed in 4% PFA for 20 min, washed repeatedly in PBST and pre-hybridized for at least 1 hr in Hyb-buffer. Digoxigenin-labeled and fluorescein-labeled RNA probes were diluted in Hyb-buffer supplemented with transfer RNA (Sigma-Aldrich) and heparin (Sigma-Aldrich), and incubated with the embryos overnight at 70˚C. After removal of the probe, embryos were washed stepwise from Hyb- to 2xSSCT, and subsequently from 0.2xSSCT to PBST. Embryos were blocked for at least 1 hr at room temperature (RT) in PBST supplemented with sheep serum and BSA before being incubated overnight at 4˚C with an anti-digoxygenin-AP antibody (1:5000; Cat: 11093274910; Roche). After removal of the antibody, embryos were washed in PBST before being transferred to TBST. The embryos were subsequently incubated in the dark on a slow rocker in dilutions of Nitro-blue tetrazolium/5-bromo-4-chloro-3-inodyl phosphate (NBT/BCIP; Cat: 11093274910; Roche) in TBST. After development of the staining, embryos were washed extensively in PBST and fixed overnight in 4% PFA at 4˚C. Before imaging, embryos were cleared in MeOH and mounted in benzylbenzoate:benzylalcohol (2:1). For two-colour

detection, after development of the NBT/BCIP staining embryos were briefly washed in PBST and 0.1 M Glycin-HCl pH = 2.2 and incubated overnight at 4°C with an anti-fluorescein antibody-AP (1:5000; Cat: 11426338910; Sigma-Aldrich). After PBST and TBST washing, ISH signal was detected with Iodonitrotetrazolium INT/BCIP (1:5000; Cat:11681460001; Sigma-Aldrich). Imaging was carried out after mounting in 100% glycerol. Cryosectioning was carried out on *tbx5a* ISH embryos previously frozen in OCT (Leica Microsystems) on dry ice at a thickness of 10 µm before slide mounting and imaging.

Accession numbers of the genes assayed by ISH: *myl7* (NM_131329), *amhc* (NM_198823), *foxa3* (NM_131299), *nppa* (NM_198800), *tbx2b* (NM_131051), *bmp4* (NM_131342), *has2* (NM_153650), *versican* (NM_001326557), and *tbx5a* (NM_130915).

### In vitro tbx5a activity assay

COS7 cells, grown in 12-well plates in DMEM supplemented with 10% FCS (Gibco-BRL) and glutamine, were transfected using polyethylenimine 25 kDa (PEI, Brunschwick) at a 1:3 ratio (DNA:PEI). Standard transfections were performed using 1.4 µg pGL3-Basic reporter vector (Promega) containing −638/+70 bp r*Nppa* promoter (reporter construct), which was co-transfected with 3 ng phRG-TK Renilla vector (Promega) as normalization control. Zebrafish *tbx5a* wild type (wt) and mutant (*hst* and *oug*) open-reading frames were cloned into a pCS2+ vector and 300 ng of each construct was transfected along with the reporter constructs and normalization control. Experiments were performed in triplo, each with hextuplicate biological replicates. Isolation of cell extracts and subsequent luciferase assays were performed 48 hr after transfection using Luciferase Assay System according to the protocol of the manufacturer (Promega). Luciferase measurements were performed using a Promega Turner Biosystems Modulus Multimode Reader luminometer. Mean luciferase activity and standard deviation were plotted as fold activation compared to the promoter-reporter plasmid. All data was statistically validated using a one-way ANOVA for all combinations.

### Imaging

In vivo phenotypic assessment and imaging was carried out on a Leica M165FC stereomicroscope or a Zeiss StemiSV6 stereomicroscope (Carl Zeiss AG, Oberkochen, Germany). Embryos were sedated if necessary with 16 mg/ml tricaine (MS222; Sigma-Aldrich) in E3 medium. ISH imaging was performed using a Zeiss Axioplan microscope (Carl Zeiss AG). Images were captured with a DFC420 digital microscope camera (Leica Microsystems). Confocal imaging was carried out on a Leica SPE or SP8 confocal microscope (Leica Microsystems). Multiphoton imaging was carried out on a Leica SP5 or SP8 confocal microscope (Leica Microsystems). Time-lapse imaging was carried out on sedated, PTU-treated, *tnnt2a* morpholino oligo-injected and dechorionated embryos mounted in 0.25% agarose in E3 medium. Images were acquired using a Leica SP5 or SP8 multiphoton microscope and stacks were acquired approximately every 10 min for about 16 hr.

Acquisition resolution of the images (x; y; z) in µm per pixel: Confocal timelapses: 0.889; 0.889; 2.000; Confocal live imaging (still): 0.604; 0.604; 1.000; Confocal fluorescent immunolabeling: 0.284; 0.284; 1.000.

### Outer and inner curvature definition

Throughout the study, we defined the inner- and outer curvatures of the chambers as the long and short contours respectively visible in the ventral view of the 48 hpf heart. In the ventricle, the outer curvature is on the left of the chamber and the inner curvature on the right, and vice-versa for the atrium. The boundary in-between the inner and outer curvatures was not defined as additional markers were not available to us.

### Image analysis

Time-lapse: Imaris software (Oxford Imaging) was used to generate time-lapse movies and automated cell tracking in 3D, followed by manual inspection of individual tracks.

Time lapse movies spanned approximately 28 hpf-38 hpf, with a frame (full stack) acquisition period of approximately 13 min. For each movie analyzed, tracks were selected if they were contained a minimum of 15 acquisition points. Drift correction was applied in Imaris prior to track analysis to correct for displacement of the whole heart during image acquisition. All data presented in the

manuscript on time-lapse movies were generated in Imaris and subsequently processed in Excel (Microsoft) if required.

Cell roundness: cell roundness assessment was carried out in Fiji freeware (https://fiji.sc/). Roundness of a cell is defined as:

Cell counting: cell counting was carried out in Volocity (Perkin Elmer) or Imaris (Oxford Imaging) on confocal-acquired 3D stacks.

Straightness Index: The straightness index is defined as the ratio between the length of a straight line from the start to the end of the left/right border at the edge on the right side of the ventricle (ventral view) and the length of the actual border as measured on the surface of the heart.

Details of the cell trajectory analyses are given in Appendix 1-Supplementary Methods.

## Statistics

Statistical assays were carried out in Graphpad Prism 9.0 (GraphPad Software). Statistical analysis for average total rotation angle, angular velocities, and twisting angle were performed with the Python packages scipy (*Virtanen et al., 2020*) and statsmodels (*Seabold and Perktold, 2010*).

*Figure 1J*: One-way ANOVA with Tukey's multiple comparison test; for all pairwise comparisons ****; p<0.0001 except *empty* vs *oug* ns; p=0.5950.

*Figure 2K*: One-way ANOVA comparing all possible combinations among ventricle, atrium, and AV canal of wild type and *oug* hearts, followed by Mann-Whitney/Wilcoxon rank-sum test and Bonferroni-correction for multiple comparison, p values and significance levels are reported in the figure panel.

*Figure 4I*: One-way ANOVA comparing all possible combinations among ventricle, atrium, and AV canal of wild type and *oug* hearts, followed by Mann-Whitney/Wilcoxon rank-sum test and Bonferroni-correction for multiple comparison, p values and significance levels are reported in the figure panel.

*Figure 4K*: Two-tailed, non-paired Student's t-test; p values and significance levels are reported in the figure panel.

*Figure 4L*: Two-tailed, non-parametric Mann-Whitney U test, p values and significance levels are reported in the figure panel.

*Figure 5B*: One-way ANOVA with Bonferroni's multiple comparison test; p values and significance levels are reported in the figure panel.

*Figure 6B*: One-way ANOVA with Bonferroni's multiple comparison test; p values and significance levels are reported in the figure panel.

*Figure 7D*: Two-tailed, non-paired Student's t-test; p values and significance levels are reported in the figure panel.

*Figure 8C*: One-way ANOVA with Tukey's multiple comparison test; p values and significance levels are reported in the figure panel.

## Data collection

*Figure 1* (C) and (H): representative pictures of a minimum of three independent experiments. Numbers of samples are reported in the figure.

(G): Number of embryos analyzed (per cross): wt x wt: n = 94; $oug^{-/-}$ x $oug^{-/-}$: n = 134; $hst^{-/-}$ x $hst^{-/-}$: n = 125; $oug^{+/-} \pm hst^{+/-}$: n = 298.

(J): six technical and biological repeats.

*Figure 2* (A–K): representative pictures and data collected on five technical and biological repeats.

*Figure 3* (A'): representative pictures of two technical and biological repeats.

(B–B'): representative pictures of six technical and biological repeats.

(C–C'): representative pictures of six technical and biological repeats.

*Figure 4* (A–I): representative pictures and data collected on five technical and biological repeats.

(J–L): data collected on five technical and biological repeats per genotype.

*Figure 5* (A): number of samples is reported in the figure panels.

(B): DMSO: nine samples; SU5402:13 samples.

(D): number of samples is reported in the figure panels.

*Figure 6* (A,B): number of samples is reported in B.

(C): number of samples is reported in the figure panels.

*Figure 7* (A): representative pictures of three biological and technical replicates per genotype.

(B): Data points: for all points $5 < n < 9$ unless *: $n = 2$.

(C–D): representative pictures and data collected on four biological and technical replicates.

*Figure 8* (A–C): representative pictures of a minimum of six biological and technical replicates, as reported in panel C.

(B): representative pictures of a minimum of five biological and technical replicates.

(D–D'''): number of biological and technical replicates are reported in the figure panels.

## Acknowledgements

The authors thank Anko de Graaff (Hubrecht Imaging Center) for assistance with microscopic imaging, Phong Nguyen (Hubrecht Institute) for carrying out the ISH cryosections and critically reading the manuscript and Hessel Honkoop (Hubrecht Institute) for critically reading the manuscript. Funding: The authors wish to acknowledge the support from the Dutch Heart Foundation grant CVON2014-18CONCOR-GENES to JB and VMC, support from the Nederlandse Organisatie voor Wetenschappelijk Onderzoek (Nederlandse Wetenschapsagenda Startimpuls) to ESC and Nederlandse Organisatie voor Wetenschappelijk Onderzoek grant NWO/ENW-VICI 865.17.004 to RMHM.

## Additional information

### Funding

| Funder | Grant reference number | Author |
|---|---|---|
| Hartstichting | CVON2014-18CONCOR-GENES | Vincent M Christoffels Jeroen Bakkers |
| Nederlandse Organisatie voor Wetenschappelijk Onderzoek | NWO/ENW-VICI 865.17.004 | Roeland MH Merks |
| Nederlandse Organisatie voor Wetenschappelijk Onderzoek | Nederlandse Wetenschapsagenda Startimpuls | Enrico Sandro Colizzi |

The funders had no role in study design, data collection and interpretation, or the decision to submit the work for publication.

### Author contributions

Federico Tessadori, Conceptualization, Formal analysis, Supervision, Validation, Investigation, Methodology, Writing - original draft; Erika Tsingos, Enrico Sandro Colizzi, Conceptualization, Formal analysis, Validation, Investigation, Methodology, Writing - review and editing; Fabian Kruse, Formal analysis, Validation, Investigation; Susanne C van den Brink, Formal analysis, Investigation; Malou van den Boogaard, Formal analysis, Investigation, Writing - original draft; Vincent M Christoffels, Supervision, Funding acquisition; Roeland MH Merks, Conceptualization, Supervision, Funding acquisition, Methodology, Writing - review and editing; Jeroen Bakkers, Conceptualization, Supervision, Funding acquisition, Methodology, Writing - original draft

### Author ORCIDs

Federico Tessadori https://orcid.org/0000-0001-9975-0546
Erika Tsingos https://orcid.org/0000-0002-7267-160X
Enrico Sandro Colizzi http://orcid.org/0000-0003-1709-4499
Susanne C van den Brink https://orcid.org/0000-0003-3683-7737
Vincent M Christoffels https://orcid.org/0000-0003-4131-2636
Jeroen Bakkers https://orcid.org/0000-0002-9418-0422

### Decision letter and Author response

Decision letter https://doi.org/10.7554/eLife.61733.sa1
Author response https://doi.org/10.7554/eLife.61733.sa2

## Additional files

### Supplementary files

- Transparent reporting form

### Data availability

Data generated during this study are included in the manuscript and supporting information.

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

## Appendix 1

### Computational unfolding of the heart axis onto a straight reference axis

Here we detail the algorithm we use to quantify the rotation of each cell during heart development. We assume that the displacement of the cells can be disentangled into three types of movement: (i) translation of the whole heart tube (*Appendix 1—figure 1 a-b*), (ii) folding of the tube over the atrio-ventricular canal (AVC) (*Appendix 1—figure 1 c*), and, finally the (iii) rotation along the AVC-atrium axis viz. the AVC-ventricle axis, which we aim to extract from the data (*Appendix 1—figure 1 d*). During translation, all cells displace by the same distance and into the same direction. During folding the whole heart tube folds over the AVC that acts a 'hinge'. We aim to remove the displacement and folding components from the cell tracking data, so that we can quantify the cells' rotation over the AVC-atrium or the AVC-ventricle axis.

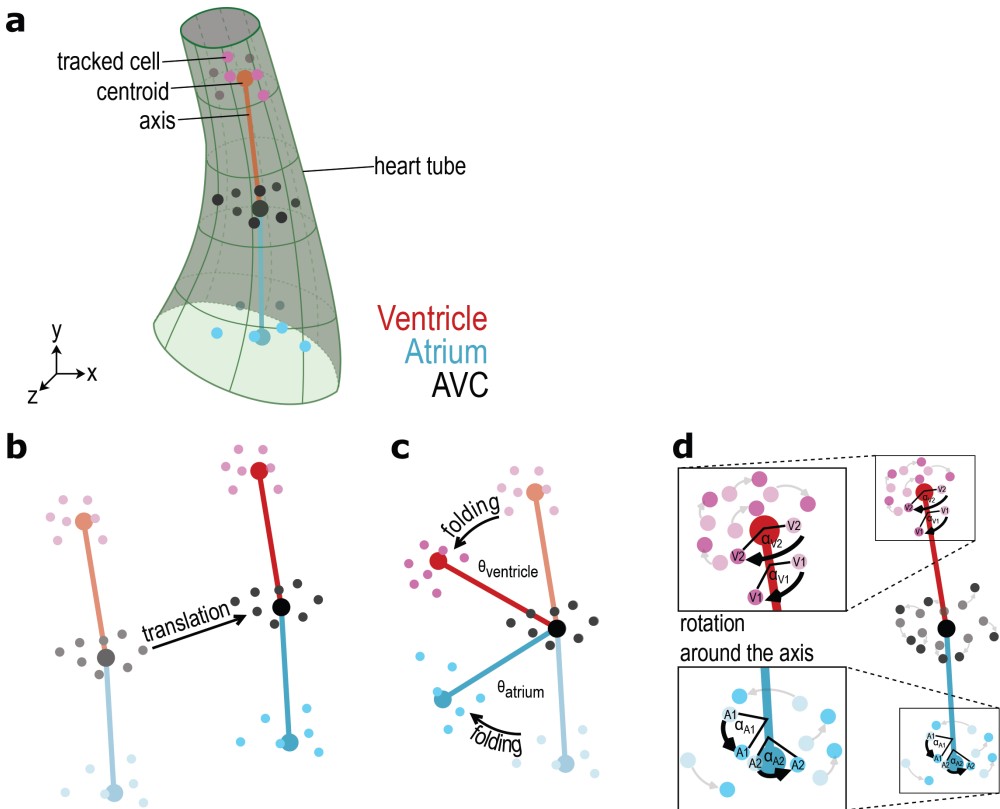

**Appendix 1—figure 1.** Scheme of the types of motion described in the text. (**a**) Scheme of the heart tube with a few cells manually marked for each category, the respective category centroid calculated from the marked cells' average position, and the axes linking these centroids. The cells are drawn in different sizes to highlight their position at the front or back of the heart tube. The centroid is larger than the cells. (**b**) Example of translation. (**c**) Example of heart tube folding over the hinge-like AVC. (**d**) Example of cell rotation around the axis of the respective category.

In short, we extract the cell positions at successive time points from the microscopy data. For each time point we translate the cell positions so that the AVC remains centered at the origin (step 1; see also *Figure 2I* in the main text). We then unfold atrium and ventricle by rotating the cells so that both the atrium-AVC axis and the ventricle-AVC axis lie onto their respective reference axes (step 2; see also *Figure 2I',I''* in the main text). Finally, we calculate the rotation of the cells by measuring the angle subtended by their displacement per time step, projected on the plane perpendicular to their axis (step 3; see also *Figure 2I'''* in the main text). In the following we explain each step in more detail.

## Step 1. Translate AVC to origin

We subdivide the cell tracks into categories belonging to atrium (A), ventricle (V), and AVC by manual annotation of the data in Imaris. In the subsequent analysis, we exclude all timepoints where any of these categories has less than 5 cells. We define the centroid $C_h(t)$ of a category $h$ as the average coordinate of the point cloud consisting of all cells in the respective category at time $t$.

In order to let the centroids of the AVC coincide throughout the time-lapse and remove translation due to drift (*Appendix 1—figure 1b*), we first translate the whole dataset such that the centroid of the AVC at each time point is located at the origin of the coordinate system. The centroid of the AVC is defined for each timepoint $t$ as the sum of each AVC cell $i$'s position $P_{i,\mathrm{AVC}}(t)$ at that timepoint divided by the total number of AVC cells at that timepoint $N_{\mathrm{AVC}}(t)$

$$C_{\mathrm{AVC}}(t) = \frac{1}{N_{\mathrm{AVC}}(t)} \sum_{i=0}^{i=N_{\mathrm{AVC}}(t)} P_{i,\mathrm{AVC}}(t).$$ (1)

We pick the AVC centroid at the earliest timepoint as a reference point

$$C_{\mathrm{AVC,r}} = C_{\mathrm{AVC}}(t_{\min})$$ (2)

and subtract this reference from each cell $i$'s position $P_i(t)$ at every timepoint $t$ to obtain a cell position vector for each cell in all the categories

$$\vec{p}_i^{\,*}(t) = P_i(t) - C_{\mathrm{AVC,r}}.$$ (3)

Then, we calculate the AVC centroids $C_{\mathrm{AVC}}^*(t)$ using the translated coordinates $\vec{p}_i^{\,*}(t)$ analogously to *Equation 19*. The displacement of $C_{\mathrm{AVC}}^*(t)$ in time quantifies the drift of the heart. We subtract each $C_{\mathrm{AVC}}^*(t)$ from all cell coordinates at the corresponding time point ($\vec{p}_i^{\,*}(t)$) to obtain

$$\vec{p}_i^{\,**}(t) = \vec{p}_i^{\,*}(t) - C_{\mathrm{AVC}}^*(t),$$ (4)

which are the coordinates of all cells at all timepoints, translated such that the AVC centroid remains located at the origin.

Similar to the AVC centroid, we define the centroid of atrium $C_{\mathrm{A}}(t)$ and ventricle $C_{\mathrm{V}}(t)$ as the average position of atrial and ventricular cells, analogously to *Equation 19*. We calculate these centroids from the translated coordinates $\vec{p}_i^{\,**}(t)$, and denote them as $C_{\mathrm{A}}^{**}(t)$ and $C_{\mathrm{V}}^{**}(t)$. For each timepoint, we can now define two axes, one for atrium $\vec{a}_{\mathrm{A}}(t)$ and one for ventricle $\vec{a}_{\mathrm{V}}(t)$, as the vector from the AVC centroid to the chamber centroid:

$$\vec{a}_{\mathrm{A}}(t) = C_{\mathrm{AVC}}^{**}(t) - C_{\mathrm{A}}^{**}(t) \text{ and } \vec{a}_{\mathrm{V}}(t) = C_{\mathrm{V}}^{**}(t) - C_{\mathrm{AVC}}^{**}(t).$$ (5)

## Step 2: Unfolding of heart axis

In order to detect twisting of the heart, we removed the folding caused by the angular movement of the axes (*Appendix 1—figure 1c*). For each of the chambers, a reference axis is defined as,

$$\vec{a}_{h,\mathrm{r}} = \vec{a}_h(t_{\min}),$$ (6)

where $h$ denotes either atrium or ventricle. That is, these are the two axes at the first timepoint of the dataset.

The datasets for all subsequent data points are then rotated such that the axes of the rotated dataset overlap with the reference axes (*Appendix 1—figure 2a-b*). To this end, we define a rotation matrix $\mathbf{R}_h(t)$ that is calculated from the unit vectors of the reference axis and the axis at timepoint $t$, respectively $a_{h,\mathrm{r}}$ and $a_h(t)$ (we denote normalised vectors with the symbol ˆ).

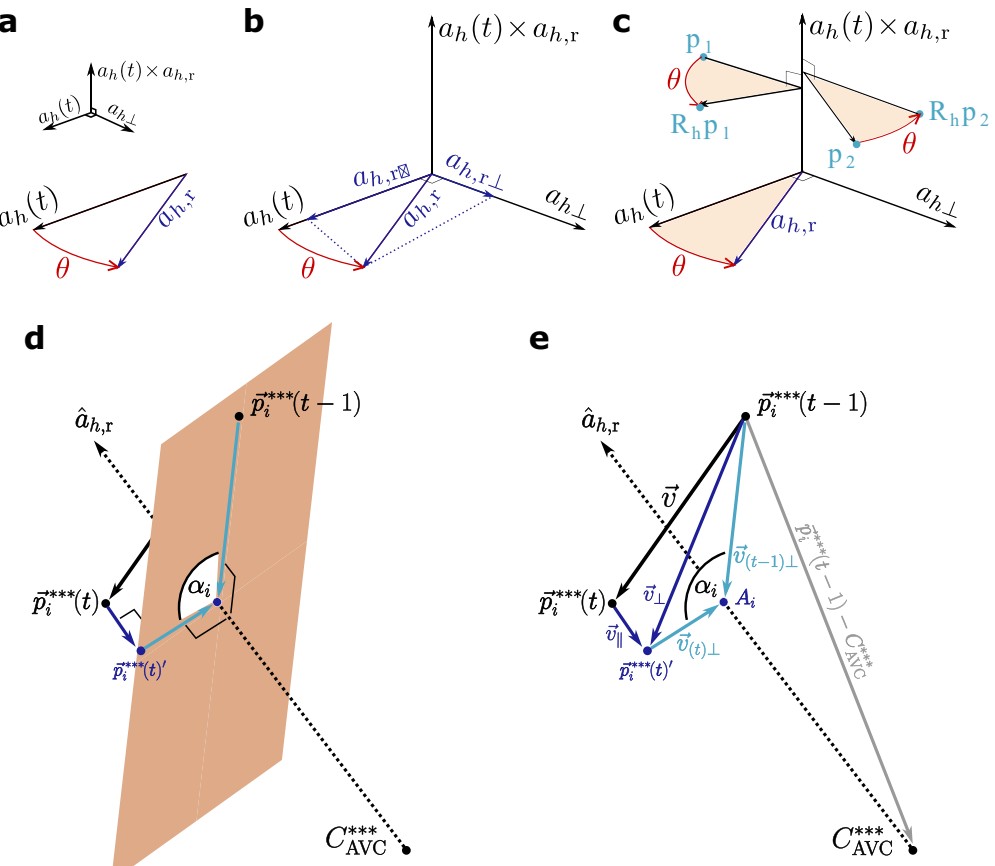

**Appendix 1—figure 2.** Scheme of the calculations. (a–b) Scheme of step 2 in the text. The folding angle $\theta$ is the angle subtended by the axis $a_h(t)$ and the reference axis $a_{h,r}$. (c) Rotating the cells by the angle defined in (a–b). (d) Scheme of the plane perpendicular to the reference axis used to calculate cell rotation around the axis. (e) Calculation of the angle of rotation of the cells around the reference axis with symbols as used in the text. Note that this is not the same angle as in (a–c).

The rotation matrix is derived from Rodrigues' rotation formula, where the axis of rotation is the normalised cross product between $a_h(t)$ and $a_{h,r}$ (*Appendix 1—figure 2a*).

$$\mathbf{R}_h(t) = I + [a]_\times + [a]_\times^2 \frac{1}{1 + \hat{a}_{h,r} \cdot \hat{a}_h(t)} \tag{7}$$

where **I** is the 3 × 3 identity matrix, and the term in square brackets represents the skew-symmetric cross product matrix

$$[a]_\times = \left[\hat{a}_h(t) \times \hat{a}_{h,r}\right]_\times = \left[-\left(\hat{a}_{h,r} \times \hat{a}_h(t)\right)\right]_\times = \begin{bmatrix} 0 & +k_z & -k_y \\ -k_z & 0 & +k_x \\ +k_y & -k_x & 0 \end{bmatrix}, \tag{8}$$

with

$$\begin{bmatrix} k_x \\ k_y \\ k_z \end{bmatrix} = \hat{a}_h(t) \times \hat{a}_{h,r}. \tag{9}$$

The rotation matrix is then multiplied to the coordinate of each cell $i$ at every timepoint $t$ of the corresponding chamber $h$ (atrium or ventricle) (*Appendix 1—figure 2c*).

$$\vec{p}_i^{***}(t) = \mathbf{R}_h(t) \cdot \vec{p}_i^{**}(t). \tag{10}$$

After this operation, the axes of the respective chambers $\vec{a}_h$ coincide at every timepoint with the reference axis, allowing us to perform all further calculations with respect to the reference unit vector axis $\hat{a}_{h,r}$. To correct the folding on AVC cells, we used the angle obtained from the above calculation using the ventricle axis as reference.

## Step 3: Calculation of cell rotation around the axis

By unfolding the heart in step 2 (i.e. rotating the axes at all timepoints onto a reference at $t_{\min}$), we have disentangled translation and folding from the rotation of a cell around its axis (*Appendix 1— figure 1 d*). We can now calculate this rotation. *Appendix 1—figure 2 d-e* illustrate the steps of the calculation. Briefly, for each cell we define a plane on which the rotation is measured (*Appendix 1— figure 2 d*). This plane is perpendicular to the axis and passes by the cell at timepoint $t - 1$. We project the cell's position at time $t$ on this plane to obtain $\vec{p}_i^{***}(t)'$. Then, we define two vectors on this plane: The first vector, $\vec{v}_{(t-1)\perp}$, spans from the plane-axis intersection $A_i$ to the position of the cell $\vec{p}_i^{***}(t-1)$, and the second vector, $\vec{v}_{t\perp}$, spans from the plane-axis intersection $A_i$ to the projected position $\vec{p}_i^{***}(t)'$. The angle $\alpha_i(t)$ between these vectors is defined as the rotation angle of the cell over that time step $t$. We neglect off-plane displacement, thus considering only the projection of the displacement on the plane itself for the rotation. The rotation angle divided by $t$ yields the rotational velocity $\omega_i(t)$.

Mathematically, for each cell $i$, we define vectors $v_i(t)$ that give the displacement of the cells between subsequent time steps

$$\vec{v}_i(t) = \vec{p}_i^{***}(t) - \vec{p}_i^{***}(t-1). \tag{11}$$

We seek to find the point $A_i$ on the axis. This point intersects the plane perpendicular to the axis that passes by $\vec{p}_i^{***}(t-1)$. It is calculated as:

$$A_i = C_{AVC}^{***} + \hat{a}_{h,r} \frac{\left(\vec{p}_i^{***}(t-1) - C_{AVC}^{***}\right) \cdot \hat{a}_{h,r}}{\hat{a}_{h,r} \cdot \hat{a}_{h,r}} \tag{12}$$

Intuitively, the point $A_i$ is the projection on the axis of the vector spanning from $\vec{p}_i^{***}(t-1)$ to $C_{AVC}^{***}$, which can be calculated as the dot product between the axis and this vector. Then, we calculate the component of the displacement vector $\vec{v}_i(t)$ parallel to the plane perpendicular to the axis and passing by $A_i$ (for clarity, we drop the subscript $i$ and time dependence when we denote the vectors $\vec{v}_i(t)$ and simply write $\vec{v}$). To this end, we decompose the displacement vectors $\vec{v}$ into components parallel to (‖) to and perpendicular to (⊥) the corresponding category's unit vector axis

$$\vec{v}_{\parallel} = \left(\vec{v} \cdot \hat{a}_{h,r}\right) \cdot \hat{a}_{h,r} \text{ and } \vec{v}_{\perp} = \vec{v} - \vec{v}_{\parallel} \tag{13}$$

We then project the cell's position in the current step $\vec{p}_i^{***}(t)$ onto the plane

$$\vec{p}_i^{***}(t)' = \vec{p}_i^{***} + \vec{v}_{\parallel}. \tag{14}$$

Now we obtain the vectors in the plane

$$\vec{v}_{(t-1)\perp} = \vec{p}_i^{***}(t-1) - A_i \text{ and } \vec{v}_{t\perp} = \vec{p}_i^{***}(t)' - A_i, \tag{15}$$

which we use to calculate the angle of rotation of the cell around the axis

$$\alpha_i(t) = \tan^{-1} \frac{\hat{a}_{h,r}\left(\vec{v}_{(t-1)\perp} \times \vec{v}_{(t)\perp}\right)}{\vec{v}_{(t-1)\perp} \cdot \vec{v}_{(t)\perp}}, \tag{16}$$

from which follows the angular velocity over the time step $t$

$$\omega_i(t) = \frac{\alpha_i(t)}{\Delta t}. \tag{17}$$

## Calculation of total rotation angle

The angles obtained in *Equation 17* are the basis for the statistical analysis presented in this article. To obtain the total rotation (or cumulative rotation) we proceed as follows: First, we average the angular velocities for each cell $i$ per category $h$ at every timepoint $t$ to obtain the mean angular velocity $\bar{\omega}_h(t)$

$$\bar{\omega}_h(t) = \frac{1}{N_h(t)} \sum_{i=0}^{i=N_h(t)} \omega_{i,h}(t), \tag{18}$$

where $N_h(t)$ is the total number of cells for category $h$ at timepoint $t$. Then, we integrate the mean angular velocity over time by taking the cumulative sum, and obtain the total rotation in each of the heart tube categories

$$\bar{\alpha}_{h,\text{tot}}(t) = \sum_{\tau=t_{\min}}^{\tau=t} \bar{\omega}_h(\tau). \tag{19}$$

## Calculation of heart twisting angle

We define the twisting angle $\xi(t)$ by the absolute difference between the total rotation obtained in *Equation 19* for the atrium and the ventricle, respectively $\hat{\alpha}_{\text{A,tot}}(t)$ and $\bar{\alpha}_{h,\text{tot}}(t_{\Delta1.5})$

$$\xi(t) = \left| \bar{\alpha}_{\text{V,tot}}(t) - \bar{\alpha}_{\text{A,tot}}(t) \right|. \tag{20}$$

### Calculation of time-averaged angular and twisting velocities

To calculate statistics on the data, we obtained time-averaged angular velocities in 1.5-hour intervals both for the total rotation angle (*Figure 2K*, *Figure 4I* in the main text) and for the twisting angle (*Figure 4L* in the main text). To this end, we split the data into $t_{\Delta1.5}$ = 1.5 hr intervals and performed a linear fit of the form $y = mt + b$, where the slope m and the intercept b are fitting parameters. The slope m of the function y fitted on the total rotation angle $\bar{\alpha}_{h,\text{tot}}(t_{\Delta1.5})$ or the twisting angle $\xi(t_{\Delta1.5})$ is then equivalent to the derivative, i.e., respectively the average angular velocity in that interval $\langle \bar{\omega}_h(t_{\Delta1.5}) \rangle$ or the average twisting velocity in that interval $\langle \psi(t_{\Delta1.5}) \rangle$.

