## [Decision Letter]

**Acceptance summary:**

With an elegant combination of cell tracking and genetic tracing, this paper reveals twisting movements in cardiac chambers during heart looping in the fish. Mutant analyses show that these movements depend on the transcription factors Tbx5a and Tbx2b, which pattern chambers and their boundary at the atrio-ventricular canal. This work thus characterises intrinsic mechanisms shaping the fish heart independently of the left determinant Nodal, a process that was previously only suspected. It provides novel insight into how the heart acquires its shape to sustain efficient blood circulation.

**Decision letter after peer review:**

Thank you for submitting your article "Twisting of the heart tube during cardiac looping is a tbx5-dependent and tissue-intrinsic process" for consideration by *eLife*. Your article has been reviewed by 3 peer reviewers, including Sigolène M Meilhac as the Reviewing Editor and Reviewer #1, and the evaluation has been overseen by Didier Stainier as the Senior Editor. The following individual involved in review of your submission has agreed to reveal their identity: (Maximilian Furthauer [Reviewer #2]).

The reviewers have discussed the reviews with one another and the Reviewing Editor has drafted this decision to help you prepare a revised submission.

While cardiac looping is essential for cardiac function, the complex morphogenetic events that govern this asymmetric process remain poorly understood. Asymmetric morphogenesis has been mainly analysed in terms of direction and downstream of left Nodal signaling. The work of Tessadori et al. now addresses the contribution of other factors to shape the heart loop. This manuscript builds upon a previous study from the same group, showing that cardiac looping is independent of the initial leftward jog of the heart that is driven by left-sided Nodal activity. A recent study from another group (Lombardo et al., 2019) suggested that rotational events occur at the level of the cardiac outflow tract. The present work substantially extends these findings by providing more evidence of intrinsic mechanisms driving looping. The authors use a number of elegant approaches to provide a 3-dimensional description of this process. The presented experimental work is generally of high quality. The combination of cell tracking and genetic tracing of left markers, including with a new 0.2Intr1spaw transgene, suggests differential movements in the ventricle and atrium. A novel allele (oug), encoding a truncated version of the transcription factor tbx5a, is analysed, showing normal gut looping, indicative of normal LR asymmetry establishment. This allele is molecularly more severe than the well-known heartstrings allele ; unlike hst mutants, in oug mutant hearts AV canal specification is expanded. Oug mutants display defective heart looping, associated with defective chamber movements. This can be rescued to some extent by further loss of tbx2b, supporting a model where Tbx5a and Tbx2b act to establish chamber and AVC boundaries to promote torsional rotation of the heart tube and shape the loop. Explant experiments and pharmacological treatments, to interfere with the tube attachment and progenitor cell ingression, do not prevent heart looping. Altogether, the present work provides important new insights into the morphogenetic events that contribute to the shaping of the zebrafish heart. However, there are important issues that should be addressed.

Essential revisions:

1. The chamber movements are interesting new observations. Yet, their analysis is currently insufficient. Although images and cell tracking have been performed in 3D, it is unclear why the quantification is flattened in 2D. Angles are treated as linear values, whereas they should be treated as circular values (see statistical comments below). The avc is proposed to act as a "fixed hinge": how are cell traces/displacement vectors in this region? In Figure 2, it seems that the movement in the ventricle is towards the posterior (or venous pole), rather than the left, and so why are the movements qualified as opposite, rather than perpendicular? In addition, vectors in the dorsal/left ventricle are not opposite, so the rationale of a rotation of the ventricle is unclear. To support the claim that authors "map cardiomyocyte behavior during cardiac looping at a single-cell level", the movement of the overall chamber should be subtracted to the cell traces. Z cell displacement in Figure 2K, 4J should be analysed in a more quantitative way (e.g. mean displacement index at the atrial/ventricular inner/outer curvature). This would allow to see whether the changes observed in oug mutants are significant.

2. As the authors themselves mention in the discussion when comparing their results to Baker et al. 2008 (which used myl7:GFP), it is essential to establish which cells are actually labelled by a transgene. Given that previous studies (e.g. Figure 1D of de Campos-Baptista et al., 2008) have shown spaw to be expressed to the left of the cardiac primordium, rather than within the cmlc2-positive cardiac disc itself, a dorsal view of the 23 somite stage cardiac disc (e.g. spaw:GFP/myl7-RFP or GFP/cmlc2 two colour in situ) should be provided to clarify whether spaw:GFP and lft2:Gal4 transgenes are indeed specifically expressed in the left heart primordium. The staining of left transgenic markers is described as dorsal at 28hpf (text and Figure 3A), and ventral at 48hpf (text and Figure 3B): please explain whether this implies a 180 degree rotation or just a general flip of the heart relative to the embryo. It is unclear why spaw-GFP cells are located in the ventral part of oug mutant ventricles (Figure 4K, 8D): in wild-type animals left-originating cells give initially rise to the dorsal part of the ventricle. Through clockwise rotation of the outflow tract, these dorsal cells are then relocated to the outer curvature of the ventricle, as shown in Figure 3B. So if no rotation occurs in tbx5a/oug, why are spaw:GFP cells found in the ventral ventricle, rather than remaining in their initial dorsal position? What is the pattern of lft2BAC in oug mutants? The legend of Figure 9 reports "expansion of the space occupied by left-originating cardiomyocytes": what is the percentage of the VV, VD, AV, AD regions labelled at different stages and in different experimental conditions? What is the degree of rotation of the pattern and does it correspond to that measured by cell tracking? Are markers of the inner/outer curvature (ex nppa) also rotating?

3. In their characterization of tbx5a/oug mutants, the authors state that cardiac looping is “defective”, but a precise description of the actual type of defect is lacking. From the picture in Figure 1C it looks as if looping occurs still in the right direction, but with reduced amplitude. Is this the only type of defect observed, or are there others (e.g. absent or inverted looping)? Numbers should be included to substantiate these claims. Jogging in oug mutants, which is stated as normal, should be shown. It is also very difficult to compare the impact of experimental conditions impairing extrinsic cues (Figure 5-6), without a quantitative analysis of cardiac looping and of the patterns of left-transgenic markers. The authors should be careful when concluding that FGF-inhibition in vivo does not inhibit heart looping. They state that they “observed robust cardiac looping in the majority of hearts”, but actually the percentage of correct looping drops from 85% (17/20, Figure 5A) to 60% (12/20). No observation of the twist is provided in this condition. A caveat of explant experiments, is that the tissue may shrink and the orientation of the sample is lost. What are the parameters of the explanted tubes (pole distance, size), and which references are used to assess patterns? Thus, it is difficult to rule out extrinsic cues of heart looping.

4. The authors suggest discrepancies between oug and previously published hst mutants (Garrity et al. 2002, Camarata et al., 2010) with respect to AVC development. It is however not clear to what extent the perceived differences may just be due to differences in the use / interpretation of different markers. For example Tessadori et al. talk about “Increased expression for the AV endocardial markers”, which appears similar to Camarata et al. talking about “loss of AV boundary restriction” of AV marker genes. The authors should provide side-by-side comparisons of the appropriate in situs (has2, bmp4, tbx2b) in oug and hst backgrounds. This would be especially critical for tbx2b, given the genetic rescue experiments, and would facilitate follow-up studies that may use either mutant to further characterize the events reported here. How does the looping phenotype compare between the two mutants?

---

## [Author Response]

Essential revisions:1. The chamber movements are interesting new observations. Yet, their analysis is currently insufficient. Although images and cell tracking have been performed in 3D, it is unclear why the quantification is flattened in 2D. Angles are treated as linear values, whereas they should be treated as circular values (see statistical comments below).

In the revised version of the manuscript, we have extensively updated our analysis of the cardiomyocyte displacement. Following the reviewers’ comments, we have removed the original analysis done in 2D and present now 3D analysis and quantification of cardiomyocyte behavior.

The 3D analysis required an entirely different approach for which a detailed description can be found in the Supplementary Materials and methods. In short, we start by stabilizing the cell-tracking data in successive timepoints by removing movements due to sample drift and folding of the heart tube, then we calculate the rotation of each cell around computationally-defined axes in the ventricle and atrium.

To stabilize the data: first, we translate the positions of all cells by a vector equal to the positional difference of the AV canal’s centroid (average position of all AV canal cells) at successive time points, thus keeping the AV canal’s centroid at a fixed location over time. Second, we get two axes for each timepoint by connecting the centroid of each chamber to the centroid of the AV canal. Third, to remove displacement due to folding of these axes, we get the angle formed by the axes at a given time point to the axis at the initial reference timepoint. Finally, we use this angle to rotate each cell’s coordinates effectively “unbuckling” the heart tube computationally.

Third, after stabilization the remaining motion is either parallel to the respective heart chamber axis or perpendicular to it (the rotation we aim to quantify). Thus, we calculate the rotation of each cell as the angle subtended by the displacement perpendicular to the axis of the respective heart chamber. For the AV canal cells, we have used the axis of the ventricle. Finally, we calculate the angular velocity of a cell at one time point as the ratio between this angle and the time between successive measurements.

The advantage of calculating angular velocities is that their analysis does not require circular statistics (because angular velocities can take any value). We updated the supplementary section with a detailed description of this method (Supplementary Methods: Computational unfolding of the heart axis onto a straight reference axis), and included several explanatory figures where appropriate (e.g. Figure 2 panel I-I’’ in the main manuscript).

The results for these new calculations on the wild type embryos are described on page 7 of the revised manuscript and shown in Figure 2I-K. They further support the conclusion of the original manuscript that the atrium and ventricle rotate in opposite directions around the AV canal, which we refer to as twisting of the heart tube.

The results for these new calculations on the *oug* mutant embryos are described on pages 9-10 of the revised manuscript and shown in Figure 4H-I. From these calculations we conclude that there is a lack of twisting in the oug heart tube.

We now also calculated the twisting angle as the absolute difference between total rotation angles of the ventricle and the atrium from 28 to 38 hpf for hearts of wild type and *oug* embryos, which are plotted in Figure 4 panels J-L. This shows that while wild type hearts have an average twisting angle of 76.0° after 9 hours of time-lapse, this is significantly reduced to 34.1° in *oug* mutant hearts.

The avc is proposed to act as a "fixed hinge": how are cell traces/displacement vectors in this region?

In the revised version of the manuscript, we have added analysis of the angular velocity of all cardiomyocytes (see above). We analyzed the AV canal cells’ rotation, by calculating their motion with respect to the ventricle axis. In some replicates AV canal cells behave more similar to the ventricle while in others more similar to the atrium (Figure 2—figure supplement 2 and Figure 4—figure supplement 1). This behaviour is expected in light of their position between the atrium and ventricle segments. On average, AV canal cells do not show a rotational preference, as expected of a hinge-like region. For the calculation of the angle of displacement of cardiomyocytes in the ventricle and the atrium, we have used the center of mass of the ventricle or the atrium together with the center of mass of the AV canal to effectively draw a reference axis for each chamber that is articulate at the AV canal. Details of the calculations and results are now presented in the supplementary methods as also described in the response to essential revision 1.

In Figure 2, it seems that the movement in the ventricle is towards the posterior (or venous pole), rather than the left, and so why are the movements qualified as opposite, rather than perpendicular? In addition, vectors in the dorsal/left ventricle are not opposite, so the rationale of a rotation of the ventricle is unclear.

In this revised manuscript we have removed the description of the vector direction in the ventricle as we now have included the calculation of rotation around the central axis for both chambers. These calculations clearly demonstrate that taking as viewpoint the outflow of the heart tube ventricular cardiomyocytes display a clockwise (negative) rotation around the central axis, while atrial cardiomyocytes display a anti-clockwise (positive) rotation (shown in Figure 2J,K).

To support the claim that authors "map cardiomyocyte behavior during cardiac looping at a single-cell level", the movement of the overall chamber should be subtracted to the cell traces.

The current analysis pipeline for the angular velocity of the cardiomyocytes removes residual drift by subtracting the movement of the AV canal from all cell traces and stabilizes the heart axis – separately for atrium and ventricle – by subtracting the rotation of each chamber around the AV canal from all cell traces.

Z cell displacement in Figure 2K, 4J should be analysed in a more quantitative way (e.g. mean displacement index at the atrial/ventricular inner/outer curvature). This would allow to see whether the changes observed in oug mutants are significant.

The revised manuscript now presents analysis in 3D, so Z displacement is quantitatively analysed in all regions of the heart in the same manner as displacement in X and Y. As correctly pointed out by the reviewers, this was a shortcoming in the original version of the paper which we are confident is now addressed. These new analysis in 3D allowed the calculation of rotation of cardiomyocytes in the chambers around the central heart axis and subsequent twisting angle, which is significantly different in wild type and *oug* mutant hearts (Figure 2; Figure 4).

2. As the authors themselves mention in the discussion when comparing their results to Baker et al. 2008 (which used myl7:GFP), it is essential to establish which cells are actually labelled by a transgene. Given that previous studies (e.g. Figure 1D of de Campos-Baptista et al., 2008) have shown spaw to be expressed to the left of the cardiac primordium, rather than within the cmlc2-positive cardiac disc itself, a dorsal view of the 23 somite stage cardiac disc (e.g. spaw:GFP/myl7-RFP or GFP/cmlc2 two colour in situ) should be provided to clarify whether spaw:GFP and lft2:Gal4 transgenes are indeed specifically expressed in the left heart primordium.

Indeed, *spaw* is expressed in the left LPM and not in the cardiac disc. Spaw is a secreted protein and as such activates gene expression in neighboring tissues. For example, Spaw induces the expression of *lefty2* in the left cardiac disc (e.g. Figure 1B of de Campos-Baptista et al., 2008 and Figure 2K and O of Noël et al. 2013 pmid:24212328). In the revised version of the manuscript, we have addressed the expression of the transgenes in the cardiac disc as follows:

– For the *tg(0.2Intr1spaw:GFP)* line: we have performed confocal imaging of the cardiac disc at 23 somite stages in *Tg*(*myl7:GalFF:UAS:RFP; 0.2Intr1spaw:GFP)* embryos. While all cardiac progenitors in the cardiac disc display RFP expression, only those located in the left side of the cardiac disc additionally express the GFP reporter. A representative picture of our observations has been added to the revised version of the manuscript as panel D in Figure 3—figure supplement 1.

– For the *tg(lft2BAC:Gal4FF)* line: due to the timing of expression of lft2 in the cardiac disc (expression starts at approximately 22 somites, just before the rapid start of the jogging process) and slow maturation time of the fluorescent protein, it is not possible to image the actual fluorescent protein reporter at the cardiac disc stage. Hence, we have carried out whole mount double-colour RNA ISH for *myl7* and *GFP* on 23 somite *tg(lft2BAC:Gal4FF; UAS:GFP)* embryos. We have added a representative picture of the result of the ISH to the revised version of the manuscript as panels A and B in Figure 3—figure supplement 2. In the cardiac disc, GFP transcripts could only be detected on the left side, conforming to the *lft2* expression pattern. Additional extracardiac GFP transcripts could also be detected at the midline, and are caused by residual Gal4FF-mediated activation of the UAS:GFP in this anatomical structure.

The staining of left transgenic markers is described as dorsal at 28hpf (text and Figure 3A), and ventral at 48hpf (text and Figure 3B): please explain whether this implies a 180 degree rotation or just a general flip of the heart relative to the embryo. It is unclear why spaw-GFP cells are located in the ventral part of oug mutant ventricles (Figure 4K, 8D): in wild-type animals left-originating cells give initially rise to the dorsal part of the ventricle. Through clockwise rotation of the outflow tract, these dorsal cells are then relocated to the outer curvature of the ventricle, as shown in Figure 3B. So if no rotation occurs in tbx5a/oug, why are spaw:GFP cells found in the ventral ventricle, rather than remaining in their initial dorsal position?

The description of the dorsal (at 28 hpf) and ventral (at 48 hpf) expression of the transgenes is due to a general flip of the heart with respect to the overall 3D geometry of the embryo and does not refer to any rotation of the heart tube. To better explain this flip we have added explanatory cartoons in Figure 2—figure supplement 1. When the heart tube is formed (between 22-28 hpf) the transgenes are expressed in the left side of the cardiac disc and form the dorsal side of the heart tube at 28 hpf due to invagination of the right side of the cardiac disc. Between 28 and 48 hpf the heart flips (venous pole moves from an anterior position to a more posterior position) and because of this flip the dorsal side of the heart tube at 28 hpf will become the ventral side at 48 hpf. Therefore, without any other rotation the *tg(0.2Intr1spaw:GFP)* cells are found in the ventral side of the heart tube in the *oug/tbx5* mutant. In wild type embryos the rotation of the ventricle relocates the *tg(0.2Intr1spaw:GFP)* cells to the outer curvature.

What is the pattern of lft2BAC in oug mutants?

In the revised version of the manuscript, we have added representative imaging of *tg(lft2BAC:Gal4FF); UAS:RFP; myl7:GFP* in a *oug* mutant at 48 hpf as panel F in Figure 3—figure supplement 2. As can be appreciated, the left-originating cardiomyocytes are located in the ventral side of the almost linear heart tube and no such RFP+ cardiomyocyte can be observed at the outer curvature of the ventricle. This result is consistent with the observation of the *tg(0.2Intr1spaw:GFP)* in *oug* mutants (shown in Figure 3C).

The legend of Figure 9 reports "expansion of the space occupied by left-originating cardiomyocytes": what is the percentage of the VV, VD, AV, AD regions labelled at different stages and in different experimental conditions? What is the degree of rotation of the pattern and does it correspond to that measured by cell tracking?

We have amended the legend of Figure 9 and replaced "expansion of the space occupied by left-originating cardiomyocytes" by “disposition” as our observations do not allow us to reach the conclusion that there is expansion (ie increase of the occupied space) of the left-originating myocardium.

To answer the second part of the reviewers’ question, we calculated the twisting angle on the dataset originally used in Figure 7C,D and present these results in Author response image 1. Here we took transversal sections at the level of the AVC including both the ventricle and atrium (dashed line in A and B), of double-labeled WT (A) and *oug* (B) hearts. On these sections we measured values L_1_ and L_2_, which correspond respectively to the perimeter of the heart (L_1_, *myl7* transgenic) and the length occupied by left-originating cardiomyocytes (L_2_, *0.2IntrSpaw* transgenic). The quotient L_2_/L_1_ gives the relative fraction of left-originating cardiomyocytes and this is then converted in fraction of a complete quadrant (360°), to which the heart section is assimilated. Since at the start of heart looping, left-originating cells occupy exactly half of such a quadrant (see Figure 2A’); i.e. 180°, this value is subtracted. The resulting angular value represents the rotation of the pattern in both the ventricle and the atrium, which we refer to as twisting angle (C). Quantification of the twisting angle for wild type (n=4) and *oug* mutants (n=4) is presented in D and shows a significantly higher value for WT (approx. 47°) than for *oug* (approx. 9°). These results are in good agreement with the new calculations based on the cell tracking.

**Author response image 1. respfig1:** Measurement of the twisting angle. Transversal sections of WT (A) and *oug* (B) are used to calculate the twisting angle θ as explained in (C). For clarity, transversal sections of the heart are assimilated to a perfect ellipsoid. Quantification and statistical comparison of the measured twisting angles for the two genotypes is presented in (D). Horizontal bars: Mean +/- SEM. Statistical test: Student’s t test; **: *p*=0.0068.

Are markers of the inner/outer curvature (ex nppa) also rotating?

We were not able to answer this question as we lack reliable markers for these regions. We analyzed *nppa* expression by ISH and with a BAC transgenic line generated in our laboratory (Honkoop et al.; PMID: 31868166). In our results (Figure 1H, Figure 8B and Figure S8), we mostly observe expression of *nppa* in the entire ventricle. In the atrium, *nppa* expression appears more restricted to the outer curvature but its expression is too weak to reliably use it as an outer curvature marker.

3. In their characterization of tbx5a/oug mutants, the authors state that cardiac looping is “defective”, but a precise description of the actual type of defect is lacking. From the picture in Figure 1C it looks as if looping occurs still in the right direction, but with reduced amplitude. Is this the only type of defect observed, or are there others (e.g. absent or inverted looping)? Numbers should be included to substantiate these claims. Jogging in oug mutants, which is stated as normal, should be shown.

Following the reviewers’ comments, we have expanded and improved our characterization of the cardiac phenotype of *tbx5a/oug* mutants. As can be seen in Figure S1 in *tbx5a/oug* mutants cardiac looping can be described as in the “correct” direction but with reduced amplitude. No inversion or absence of cardiac looping was observed in *oug/tbx5* mutants. As suggested we have also added numbers to the panels in Figure 1C.

We have also added panels (including numbers) illustrating our analysis of cardiac jogging in *tbx5a/oug* mutants, which is not affected (Figure 1C in revised version). This is in line with the observation of normal left-right patterning in *oug/tbx5a* mutants.

It is also very difficult to compare the impact of experimental conditions impairing extrinsic cues (Figure 5-6), without a quantitative analysis of cardiac looping and of the patterns of left-transgenic markers. The authors should be careful when concluding that FGF-inhibition in vivo does not inhibit heart looping. They state that they “observed robust cardiac looping in the majority of hearts”, but actually the percentage of correct looping drops from 85% (17/20, Figure 5A) to 60% (12/20). No observation of the twist is provided in this condition.

To address the reviewers’ comments, we have carried out several additional experiments, the results of which are described below and reported in the revised Figure 5 and Figure 5—figure supplement 1.

– Cardiomyocyte counting at 48 hpf was carried out to confirm the effectiveness of SU5402 treatment. The results are presented in Figure S7.

– To address twisting of the heart tube, Imaging of *Tg*(*myl7:GalFF:UAS:RFP; 0.2Intr1spaw:GFP)* hearts after treatment with DMSO or SU5402 was carried out. Basing ourselves on the results (Figure 5A), we conclude that twisting still occurs after SU5402 treatment.

– To provide a better quantification of looping we measured the looping angle in SU5402-treated embryos as originally done in Figure 8. Results are shown in (Figure 5B) and statistical testing revealed no significant differences in looping angle between DMSO and SU5402 treated hearts. We removed the more qualitative descriptions of “correct” heart looping in DMSO and SU5402 treated embryos from the manuscript.

The results of the above-mentioned new experiments expand and confirm our initial conclusion that FGF inhibition does not significantly impact the intrinsic ability of the heart tube to loop.

A caveat of explant experiments, is that the tissue may shrink and the orientation of the sample is lost. What are the parameters of the explanted tubes (pole distance, size), and which references are used to assess patterns? Thus, it is difficult to rule out extrinsic cues of heart looping.

We understand the concerns of the reviewers, and we do not rule out all extrinsic cues of heart looping. Our analysis and conclusion are restricted to the cell addition from the SHF, for which we find no evidence that this contributes to heart looping in zebrafish embryos. We are confident that the experiments allow us to reach the current conclusions, for a number of reasons:

– Once explanted, the heart tubes are placed in liquid culture medium without any shaking or gradient of any kind (as described in Noël et al., 2013 PMID: 24212328).

– Tissue shrinking is minimal, if present, as indicated by the scale bars added in the revised version of the manuscript (compare with “normal” 48 hpf hearts).

– The explanted hearts are derived from *tg(lft2BAC:Gal4FF; UAS:RFP; myl7:GFP)* embryos that completed left cardiac jogging. Hence the transgenes are used to relate the hearts to their original orientation in the embryo.

Based on these results and on the SU5402 treatments we conclude that addition of cells from the second heart field is not essential for heart looping morphogenesis of the zebrafish heart. The revised manuscript has been modified accordingly.

4. The authors suggest discrepancies between oug and previously published hst mutants (Garrity et al. 2002, Camarata et al., 2010) with respect to AVC development. It is however not clear to what extent the perceived differences may just be due to differences in the use / interpretation of different markers. For example Tessadori et al. talk about “Increased expression for the AV endocardial markers”, which appears similar to Camarata et al. talking about “loss of AV boundary restriction” of AV marker genes. The authors should provide side-by-side comparisons of the appropriate in situs (has2, bmp4, tbx2b) in oug and hst backgrounds. This would be especially critical for tbx2b, given the genetic rescue experiments, and would facilitate follow-up studies that may use either mutant to further characterize the events reported here.

The main difference we describe between the *hst/tbx5a* allele and the *oug/tbx5a* allele is related to the reported lack of *tbx2b* expression in the AVC of *hst/tbx5a* mutants and the expansion of *tbx2*b expression in the *oug/tbx5a* mutants. We agree with the reviewers that it is hard to make conclusions without a side-by-side comparison of the two different mutant alleles for *tbx5a*. Therefore, we have now performed RNA ISH in *hst* embryos for *has2, bmp4* and *tbx2b* and show the results side-by-side to the results on *oug/tbx5a* (Figure 1H). In contrast to the reported lack of *tbx2b* expression by Camarata et al., we observed some expression of *tbx2b* in the AVC of *hst/tbx5a* mutant embryos, however this expression was strongly reduced compared to wild type and *oug/tbx5a* mutant hearts. Based on these results we conclude that *hst* and *oug* have different effects on *tbx2b* expression in the AVC. We have adjusted the text in the Results section which now reads:

“As *tbx5a* is expressed throughout the myocardium (Figure 1H,I), where it regulates patterning of the heart in chamber (working) and AV canal (non-working) myocardium we performed in situ hybridization (ISH) using markers for the AV canal and chamber myocardium. […] In accordance with our observations on the AV canal myocardium, we also detected increased expression of the AV endocardial markers *has2* and v*ersican* (Figure 1H). ”

How does the looping phenotype compare between the two mutants?

We find that the looping phenotype is more homogeneous in *oug* than it is in *hst*. To substantiate this observation, we have provided in Figure 1—figure supplement 1 side-by-side comparison of a number of representative *myl7* ISH pictures for 2dpf hearts in *oug* and *hst.* While for *oug* we consistently observe very reduced dextral looping, for *hst* the spectrum of observed phenotype is much broader, ranging from almost wt-like robust dextral looping to moderate sinistral looping. This result is consistent with our conclusion that *hst* and *oug* have different effects on Tbx5 activity.